# Genome-wide mapping of native co-localized G4s and R-loops in living cells

Ting Liu[1†], Xing Shen[1†], Yijia Ren[1†], Hongyu Lu[1], Yu Liu[2], Chong Chen[2], Lin Yu[1]*, Zhihong Xue[1,3]*

[1]Key Laboratory of Birth Defects and Related Disease of Women and Children of MOE, Department of Pediatrics, West China Second University Hospital, State Key Laboratory of Biotherapy and Collaborative Innovation Center of Biotherapy, Sichuan University, Chengdu, China; [2]Department of Hematology and Institute of Hematology, State Key Laboratory of Biotherapy and Cancer Center, West China Hospital, Sichuan University, Chengdu, China; [3]Development and Related Diseases of Women and Children Key Laboratory of Sichuan Province, Chengdu, China

*For correspondence:
yu_scu@scu.edu.cn (LY);
xuez@scu.edu.cn (ZX)

[†]These authors contributed equally to this work

**Competing interest:** The authors declare that no competing interests exist.

**Abstract** The interplay between G4s and R-loops are emerging in regulating DNA repair, replication, and transcription. A comprehensive picture of native co-localized G4s and R-loops in living cells is currently lacking. Here, we describe the development of HepG4-seq and an optimized HBD-seq methods, which robustly capture native G4s and R-loops, respectively, in living cells. We successfully employed these methods to establish comprehensive maps of native co-localized G4s and R-loops in human HEK293 cells and mouse embryonic stem cells (mESCs). We discovered that co-localized G4s and R-loops are dynamically altered in a cell type-dependent manner and are largely localized at active promoters and enhancers of transcriptional active genes. We further demonstrated the helicase Dhx9 as a direct and major regulator that modulates the formation and resolution of co-localized G4s and R-loops. Depletion of Dhx9 impaired the self-renewal and differentiation capacities of mESCs by altering the transcription of co-localized G4s and R-loops -associated genes. Taken together, our work established that the endogenous co-localized G4s and R-loops are prevalently persisted in the regulatory regions of active genes and are involved in the transcriptional regulation of their linked genes, opening the door for exploring broader roles of co-localized G4s and R-loops in development and disease.

## eLife assessment

This study describes a **useful** antibody-free method to map both G-quadruplexes and R-loops in vertebrate cells independently of the BG4 and S9.6 antibodies. It also reveals that the helicase Dhx9 can affect the self-renewal and differentiation capacities of mESCs, perhaps by regulating co-localized G4s and R-loops. The datasets provided might constitute a good starting point for future functional studies, and although the strength of the evidence that DHX9 interferes with the ability of mESCs to differentiate by regulating directly the stability of either G4s or R-loops has been improved compared to a previous version, it is still **incomplete**.

## Introduction

Genomic DNA can form various types of non-B secondary structures, including G-quadruplexes (G4s), R-loops, Z-DNA, i-motifs, Cruciform, and others (*Matos-Rodrigues et al., 2023*). Among them, G4s and R-loops are two of the best studied. G4s are built by stacked guanine tetrads connected via Hoogsteen hydrogen bonds and can be formed by intra- or inter-molecular folding of the tetramers

(*Panyutin et al., 1989*; *Sen and Gilbert, 1988*; *Sundquist and Klug, 1989*; *Williamson et al., 1989*). R-loops are three-stranded structures containing a DNA-RNA hybrid and a displaced single-stranded DNA (*Aguilera and García-Muse, 2012*; *Xu and Clayton, 1996*). Both G4s and R-loops are involved in key biological processes, including transcription, replication, genomic instability, class switch recombination in B cells, DNA damage and repair, and telomere maintenance (*García-Muse and Aguilera, 2019*; *Hänsel-Hertsch et al., 2017*; *Varshney et al., 2020*; *Yang et al., 2023*).

R-loops appear to have a strong sequence preference with high G/C ratios (*Ginno et al., 2013*; *Ginno et al., 2012*). Reports about the interplay between R-loops and G4s are emerging. Specific G4 ligands stabilized G4s and simultaneously increase R-loop levels within minutes in human cancer cells, which finally induced DNA damage (*De Magis et al., 2019*). Reactive oxygen species (ROS) have been reported to induce G4 and R-loop formation at transcriptionally active sites, and their inter-regulation is essential for the DNA repair (*Tan et al., 2020*). In a reconstituted eukaryotic DNA replication system, the interplay of R-loops and G4s was shown to impact replication fork progression by inducing fork stalling (*Kumar et al., 2021*). Single-molecule fluorescence studies showed the existence of a positive feedback mechanism of G4 and R-loop formation during transcription, where the transcription-induced R-loop precedes and facilitates G4 formation in the non-template strand, and in turn, G4 promotes the R-loop formation in the following rounds of transcription (*Lee et al., 2020*; *Lim and Hohng, 2020*). Wulfridge et al. reported that the architectural protein CCCTC binding factor (CTCF)-bound sites are enriched for R-loops and G4s which facilitate CTCF binding to promote chromatin looping interactions (*Wulfridge et al., 2023*).

Detection of G4s and R-loops have been largely based on the use of a single-chain variable fragment (scFv) BG4 for G4s and a monoclonal antibody S9.6 for R-loops. In recent years, these two antibodies have been coupled with deep sequencing to genome-widely detect G4s and R-loops (*Galli et al., 2022*; *Ginno et al., 2012*; *Hänsel-Hertsch et al., 2016*; *Jiang et al., 2023*; *Lyu et al., 2021*). Using BG4 and S9.6 -based CUT & Tag, G4s, and R-loops showed a high degree of co-occurrence in mESCs (*Lyu et al., 2021*). However, given that a group of helicases, RNA-binding factors, endonucleases, and DNA topoisomerases cooperate to actively dissolve G4s and R-loops restoring B-formed DNA duplexes (*Hänsel-Hertsch et al., 2017*; *Varshney et al., 2020*; *Yang et al., 2023*), a steady state equilibrium is generally set at low levels in living cells under physiological conditions (*Miglietta et al., 2020*) and thus the addition of high-affinity antibodies may pull the equilibrium towards folded states. Additionally, the specificity of the S9.6 antibody on R-loops has been questioned recently for accurate quantification and mapping of R-loops (*Hartono et al., 2018*; *König et al., 2017*; *Phillips et al., 2013*).

To understand the co-localized G4s and R-loops in living cells under physiological conditions, we sought to develop an in vivo strategy for G4 profiling based on the G4-hemin complex-induced proximal labeling and R-loop profiling based on the N-terminal hybrid-binding domain (HBD) of RNase H1. Recent studies showed that G4s could tightly form a complex with the cellular cofactor hemin both in vitro and in living cells, where hemin binds by end-stacking on the terminal G-quartets of G4s without affecting the folding of G4s (*Gray et al., 2019*; *Stadlbauer et al., 2021*). The G4-hemin complex has been shown to act as a peroxidase to catalyze oxidation reactions in the presence of hydrogen peroxide ($H_2O_2$) (*Cheng et al., 2009*; *Lat et al., 2020*; *Yang et al., 2011*). The $H_2O_2$-activated G4-hemin complex oxidizes the biotin tyramide to phenoxyl redicals that covalently conjugate biotin to G4 itself and its proximal DNA within 10 nm (equivalent to approximately 31 bp) in vitro and in vivo (*Einarson and Sen, 2017*; *Lat et al., 2020*). Here, we have utilized the G4-hemin-mediated proximal biotinylation rection to develop a new method HepG4-seq (for high throughput **seq**uencing of **he**min-induced **p**roximal labeled **G4**s) to map the genomic native G4s under physiological conditions. The HBD domain of RNase H1 mediates the specific recognition of DNA/RNA hybrid in a sequence-independent manner, which is a gold standard for R-loop recognition in the cell (*Nowotny et al., 2008*). The catalytically inactive RNase H1 or its HBD domain -based methods have been successfully used to identify genome-wide native R-loops (*Chen et al., 2017*; *Wang et al., 2021*). We have adapted the 'GST-His6-2xHBD'-mediated CUT&Tag protocol (*Wang et al., 2021*) to develop the HBD-seq protocol in this study.

We have combined the HepG4-seq and HBD-seq to profile the genome-wide native co-localized G4s and R-loops with high signal-to-noise ratios in HEK293 cells and mESCs. We observed that the co-localized G4s and R-loops exhibit cell type-dependent distributions and are largely localized at

active promoters and enhancers of transcriptionally active genes. We further showed that ~70% of the co-localized G4s and R-loops in mESCs were directly bound by the helicase Dhx9 and that depletion of Dhx9 significantly altered the levels of ~6200 co-localized G4s and R-loops bound by Dhx9. Furthermore, depletion of Dhx9 was shown to impair the self-renewal and differentiation capacities of mESCs by altering the transcription of co-localized G4s and R-loops -associated genes.

## Results

### Mapping of the native DNA G4 through the G4-hemin-mediated proximal biotinylation

The DNA G4-hemin complex could act as a mimic peroxidase to oxidize the biotin tyramide to phenoxyl radicals that can covalently conjugate biotin to G4 itself and its proximal DNA within ~30 bp in the presence of $H_2O_2$ (*Cheng et al., 2009*; *Einarson and Sen, 2017*; *Lat et al., 2020*). However, the efficiency of peroxidase-mediated biotinylation on DNA is limited using the substrate biotin tyramide (*Zhou et al., 2019a*). Recently, biotin aniline (Bio-An) has been shown to have superior labeling efficiency on DNA than biotin tyramide, when catalyzed by the engineered peroxidase APEX2 (*Zhou et al., 2019a*). The free heme concentration in normal human erythrocytes is 21±2 μM (*Aich et al., 2015*). To explore the G4-hemin-mediated biotinylation in the living cells, we treated HEK293 cells with 25 μM hemin and 500 μM Bio-An for 2 hr prior to activation with 1 mM $H_2O_2$ for 1 min, and then quenched the labeling reaction and performed the immunofluorescence staining using Alexa Fluor 647 conjugated recombinant streptavidin (Strep-647) that specifically recognizes biotin. As shown in *Figure 1A*, cells treated with hemin and Bio-An exhibited a robust fluorescence signal, while the absence of either hemin or Bio-An almost completely abolished the biotinylation signals, suggesting a specific and active biotinylation activity. To understand whether the addition of hemin disturbs the formation of G4s, we performed the BG4 CUT&Tag-seq using the recombinant BG4 and Tn5 on HEK293 cells treated with and without 25 μM hemin (*Figure 1—figure supplement 1A, B*). The heatmap and profile plot analysis showed similar BG4 CUT&Tag signals between the hemin-treated and control samples (*Figure 1—figure supplement 1B*). There were only 174 BG4 CUT&Tag peaks with significantly differential signals between the hemin-treated and control samples (*Figure 1—figure supplement 1C*). These data suggest that the hemin treatment condition we used does not significantly affect G4 folding. Therefore, hemin-induced proximal biotinylation of G4s could be utilized to mark the native G4s in living cells.

The recombinant streptavidin monomer (mSA) combines the streptavidin and rhizavidin sequences to achieve specific monovalent detection of biotin or biotinylated molecules with a high affinity (Kd = 2.8 nM) (*Lim et al., 2013*). The Moon-tag system consists of a 15-amino acid peptide GP41-tag and a 123-amino acid anti-GP41-tag nanobody with an affinity of ~30 nM in vitro (*Boersma et al., 2019*). We expressed and purified the recombinant mSA fused with the anti-GP41 nanobody (mSA-scFv), and the recombinant Tn5 fused with the GP41-tag and protein G (GP41-pG-Tn5) from *E. coli* (*Figure 1—figure supplement 1A*). To map the hemin-induced biotinylated G4s, we developed a new method HepG4-seq (*Figure 1B*), where mSA-scFv recognizes the biotinylated G4s and recruits the transposase Tn5 to achieve 'Cleavage Under Targets and Tagmentation (CUT&Tag).' Deep sequencing analysis of the biotinylated G4s fragments identified 6,799 consensus peaks from two independent biological repeats in HEK293 cells, where the signals were dramatically diminished in HEK293 cells without treatment with hemin and Bio-An, suggesting the specificity of HepG4-seq (*Figure 1C*, *Supplementary file 1*). Several representative HepG4s-seq-identified G4 peaks were shown in *Figure 1D*. Genomic distribution analysis showed that G4s are mainly localized in promoters (38.7%) and gene bodies (47.1%) (*Figure 1E*).

We also evaluated the HepG4-seq-identified peaks using a G4-forming sequences (PQS) predication tool pqsfinder which has been shown to have 96% accuracy on ~400 known and experimentally observed G4 structures (*Hon et al., 2017*). The peaks identified by HepG4-seq overlap quite well with the center of pqsfinder max Scores that report the PQS quality (*Figure 1C and D*). The motif enrichment analysis by HOMER (*Heinz et al., 2010*) revealed a high prevalence of G-rich sequences in HepG4-seq peaks (*Figure 1F*). All above further validated the specificity of HepG4-seq in capturing G4s.

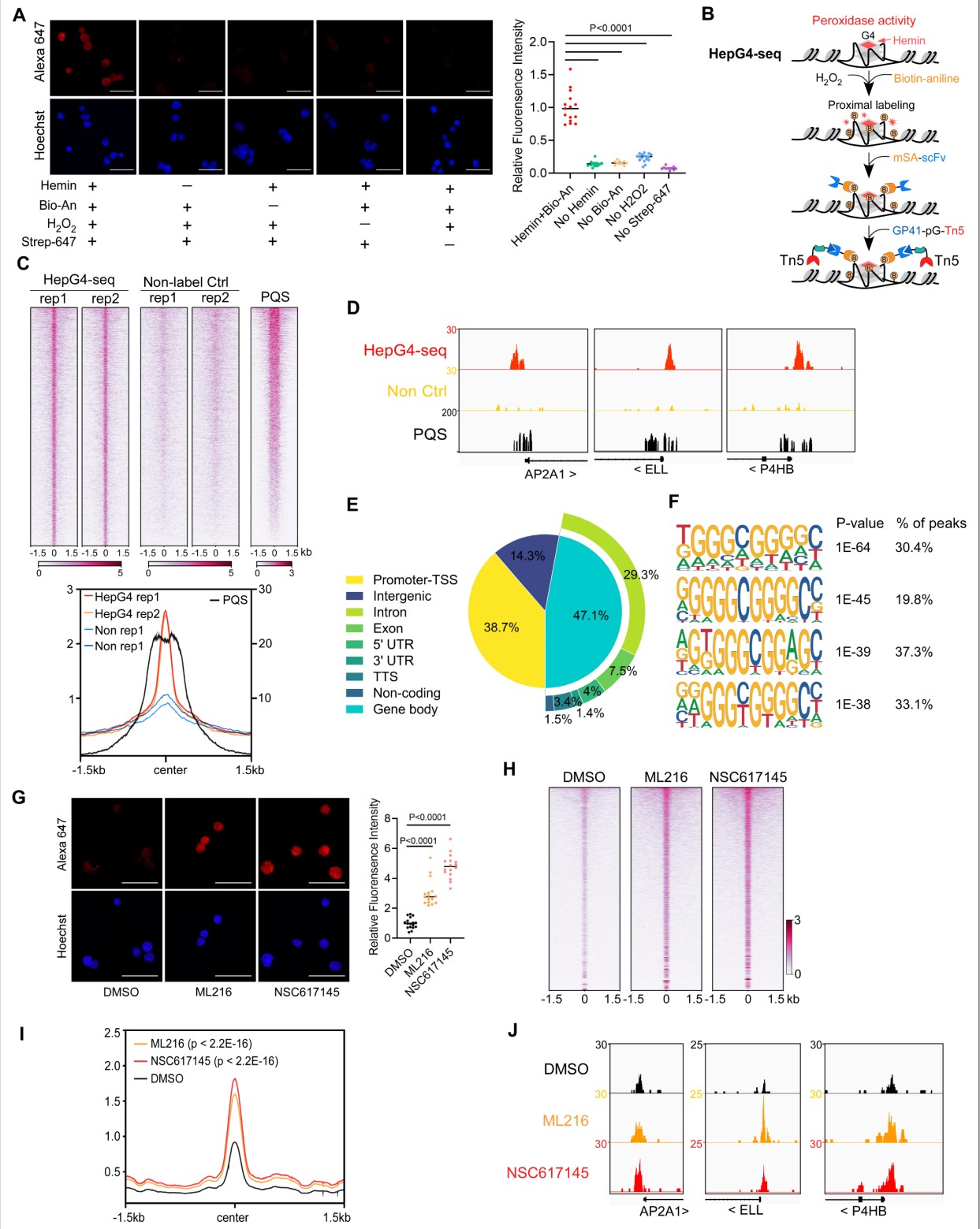

**Figure 1.** Mapping of the native DNA G-quadruplex (G4) through the G4-hemin-mediated proximal biotinylation. (**A**) Immunofluorescence staining of the HEK293 cells treated with indicated conditions using the Alexa Fluor 647 labeled recombinant streptavidin (Strep-647). Nuclei were stained with the Hoechst33342. Scale bar, 50 μm. Bio-An, Biotin-aniline; $H_2O_2$, Hydrogen peroxide. The quantified relative fluorescence intensities were shown in the right panel. This experiment was repeated once with similar results. (**B**) Schematic of the HepG4-seq procedure. SA-scFv, the recombinant fusion protein of mSA and anti-GP41 Single Chain Fragment Variable (scFv); GP41-pG-Tn5, the recombinant fusion protein of the GP41 tag, protein G and Tn5. (**C**) Top:

*Figure 1 continued on next page*

*Figure 1 continued*

Heatmap showing the signal of BG4-seq, HepG4-seq, and max Scores of PQS ±1.5 kb around the center of peaks identified by HepG4-seq in HEK293 cells. Color scales represent the density of the signals. Bottom: Profile plot showing the average signal of HepG4-seq reads ±1.5 kb around the center of peaks and the average maxScores of PQS calculated by pqsfinder at the same positions. HepG4 rep1/rep2, two biologically independent HepG4-seq replicates in HEK293 cells; Non rep1/rep2, two biologically independent non-label negative control replicates in HEK293 cells. (**D**) Representative genome browser tracks showing HepG4-seq (red), non-label negative control of HepG4-seq (yellow), and PQS (black) signals in HEK293 cells along the indicated genomic loci. (**E**) Distribution of HepG4-seq signals in HEK293 cells in different gene features. (**F**) The top enriched motifs on the HepG4-seq peaks in HEK293 cells. (**G**) Immunofluorescence staining of the HEK293 cells treated with DMSO, ML216 (25 μM), or NSC617145 (3 μM) using the Alexa Fluor 647 labeled recombinant streptavidin. Nuclei were stained with the Hoechst33342. Scale bar, 50 μm. The quantified relative fluorescence intensities were shown in the right panel. This experiment was repeated once with similar results. (**H**) Heatmap showing the HepG4-seq signals ±1.5 kb around the center of peaks identified in HEK293 cells treated with DMSO, ML216 (25 μM), or NSC617145 (3 μM). Color scales represent the density of the signals. (**I**) Profile plot showing the average signal of HepG4-seq reads ±1.5 kb around the center of peaks identified in HEK293 cells treated with DMSO, ML216 (25 μM), or NSC617145 (3 μM). The p values (ML216 v.s. DMSO; NSC617145 v.s. DMSO) were calculated using the Mann-Whitney test. (**J**) Representative genome browser tracks showing the HepG4-seq signals in HEK293 cells treated with DMSO, ML216 (25 μM), or NSC617145 (3 μM) along the indicated genomic loci.

The online version of this article includes the following source data and figure supplement(s) for figure 1:

**Source data 1.** PDF file containing original microscope images displayed in *Figure 1A and G*, indicating the selected regions.

**Source data 2.** Original files of microscope images are displayed in *Figure 1A and G*.

**Figure supplement 1.** Comparison between the BG4-seq and HepG4-seq.

**Figure supplement 1—source data 1.** PDF file containing original gel pictures for *Figure 1—figure supplement 1A*, indicating the selected regions.

**Figure supplement 1—source data 2.** Original files for gel pictures are displayed in *Figure 1—figure supplement 1A*.

## Induction of DNA G4s by inhibiting G4 resolving helicase

The RecQ-like helicases Bloom syndrome protein (BLM) and Werner syndrome ATP-dependent helicase (WRN) are the first recognized and the best-characterized DNA G4-resolving mammalian helicases (*Fry and Loeb, 1999*; *Mendoza et al., 2016*; *Mohaghegh et al., 2001*). The small molecules ML216 and NSC617145 are selective and cell-permeable inhibitors of BLM and WRN, respectively, by inhibiting their ATPase activity (*Aggarwal et al., 2013a*; *Aggarwal et al., 2013b*; *Nguyen et al., 2013*). To investigate the effect of BLM or WRN inhibition on native G4s, we treated HEK293 cells with ML216 or NSC617145 for 16 hr and then labeled cellular G4s by hemin-G4-induced biotinylation in living cells. Immunofluorescence staining of the treated cells using Strep-647 showed that the treatment of ML216 or NSC617145 remarkably elevated signals of native G4s (*Figure 1G*). Furthermore, we performed HepG4-seq on HEK293 cells treated with ML216 or NSC617145. Notably, HepG4-seq identified 77,003 peaks from ML216- or NSC617145- treated HEK293 cells, and ~70,000 new G4 peaks were induced by inhibition of BLM or WRN (*Supplementary file 1*). The signals of G4s detected by HepG4-seq were significantly increased after inhibiting BLM or WRN (*Figure 1H and I*) (Mann-Whitney test, p<2.2E-16). Representative G4s peaks are shown in *Figure 1J*. In addition, the top enriched motifs in the 70000 extra HepG4-seq peaks are G-riched (*Figure 1—figure supplement 1E*). Taken together, these data suggest that HepG4-seq is able to efficiently detect dynamic native G4s.

## Mapping of native co-localized G4s and R-loops in HEK293 cells

The HBD domain of RNase H1 has been demonstrated as a DNA/RNA hybrid recognition sensor and applied to identify genome-wide native DNA/RNA hybrids using the recombinant GST-His6-2xHBD coupling with Tn5-based CUT&Tag (*Nowotny et al., 2008*; *Wang et al., 2021*). Given that the GST-fusion proteins are prone to form variable high molecular-weight aggregates and these aggregates often undermine the reliability of the fusion proteins (*Deceglie et al., 2014*; *Ki and Pack, 2020*), we produced the recombinant two copies of HBDs fused with EGFP and V5-tag (HBD-V5) (*Figure 1—figure supplement 1A*) and used the anti-V5 tag antibody instead of the anti-His tag antibody for the CUT&Tag-seq. We call this modified protocol as HBD-seq for mapping the native R-loops in cells (*Figure 2A*). We performed the HBD-seq on HEK293 cells, and revealed 42,488 consensus native R-loops peaks with a high signal-to-noise ratio while the HBD-seq signals were dramatically diminished in HEK293 cells treated with the RNases prior to HBD-seq (*Figure 2B*), suggesting the specificity of HBD-seq in detecting native R-loops.

We then analyzed the regions co-occupied by both HepG4-seq-identified G4s and HBD-seq-identified R-loops, and revealed 5030 native co-localized peaks in HEK293 cells, ranging in size from

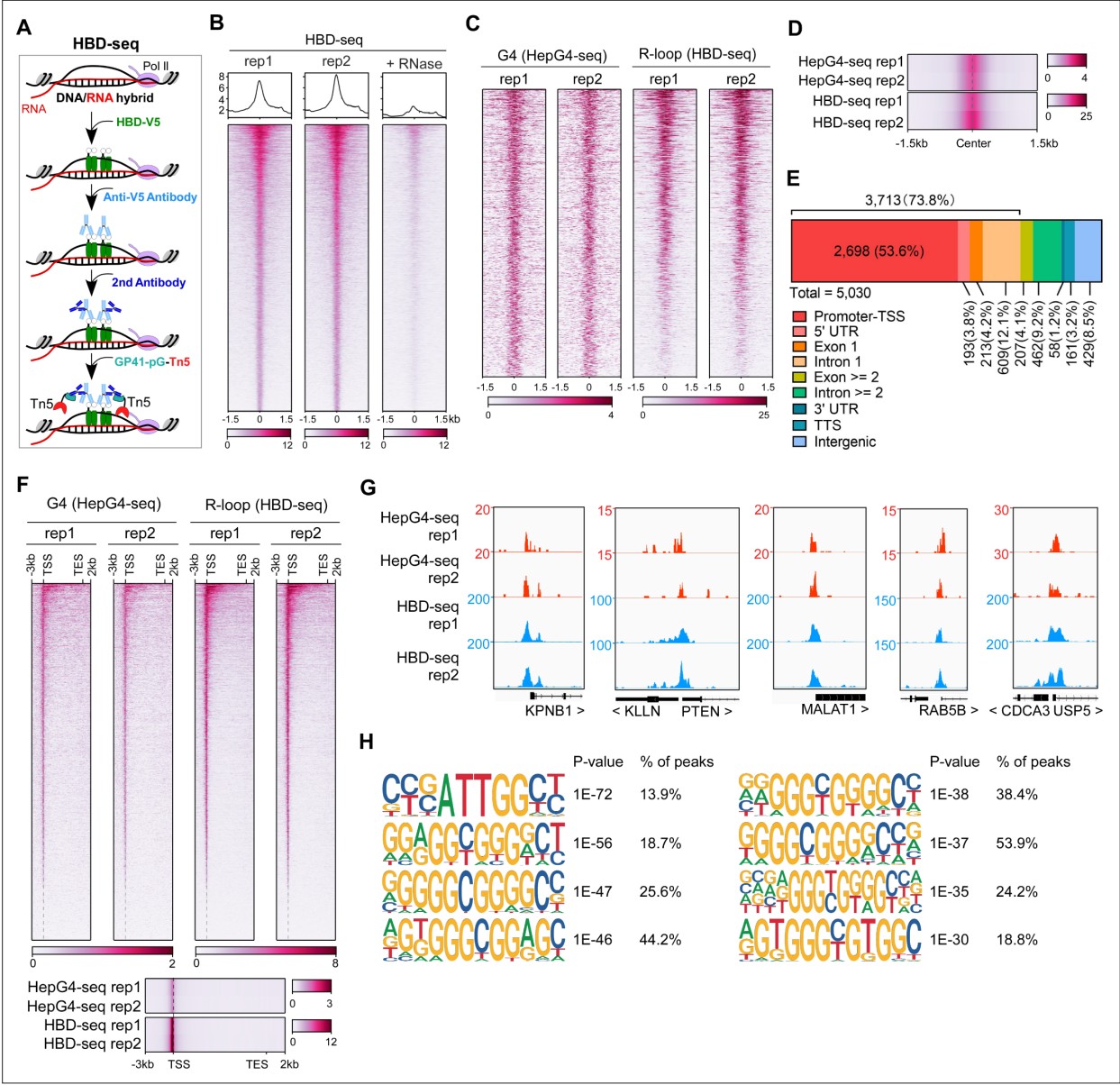

**Figure 2.** Mapping of the co-localized G-quadruplex (G4) and R-loop in the HEK 293 cells by combining the HepG4-seq and hybrid-binding domain (HBD)-seq. (**A**) Schematic of the HBD-seq procedure. HBD-V5, the recombinant fusion protein of the N-terminal hybrid-binding domain (HBD) of RNase H1 and V5 tag; GP41-pG-Tn5, the recombinant fusion protein of the GP41 tag, protein G, and Tn5. (**B**) Heatmap showing the signal of HBD-seq reads ±1.5 kb around the center of peaks in HEK293 cells. Two biologically independent replicates are shown. '+RNase' represents the treatment of RNase A and RNase H prior to the HBD-seq. Color scales represent the density of the signals. (**C**) Heatmap showing the signal of HepG4-seq and HBD-seq reads ±1.5 kb around the center of co-localized G4 and R-loop peaks in HEK293 cells. Color scales represent the density of the signals. (**D**) Profile plot showing the average signal of HepG4-seq and HBD-seq reads ±1.5 kb around the center of co-localized G4 and R-loop peaks in HEK293 cells. The plot is visualized using the heatmap. Color scales represent the density of the signals. (**E**) Distribution of the co-localized G4s and R-loops in HEK293 cells in different gene features. (**F**) Top: Heatmap showing the signal of HepG4-seq and HBD-seq reads of the co-localized G4s and R-loops in HEK293 cells along the gene body, 3 kb upstream of transcription start site (TSS) and 2 kb downstream of transcription end site (TES). Bottom: Profile plot showing the average signal of HepG4-seq and HBD-seq reads of the co-localized G4s and R-loops in HEK293 cells along the indicated gene features. The plot is visualized using the heatmap. Color scales represent the density of the signals. (**G**) Representative genome browser tracks showing the HepG4-seq and HBD-seq signals of the co-localized G4s and R-loops in HEK293 cells along the indicated genomic loci. (**H**) The top enriched motifs of the co-localized G4s and R-loops in HEK293 cells.

100 bp to ~1.5 kb (*Figure 2C and D*, *Figure 1—figure supplement 1D*). 73.8% of these co-localized peaks are localized at promoters, 5'UTR, exon1, and intron 1 (*Figure 2E*). When we performed a metagene analysis of these co-localized peaks, a distinct peak was detected around the transcription start site (TSS) (*Figure 2F*). Representative co-localized peaks are shown in the *Figure 2G*. The motifs enrichment analysis by HOMER (*Heinz et al., 2010*) showed that G-rich sequences are highly enriched in the co-localized peaks (*Figure 2H*).

## The co-localized G4s and R-loops-mediated transcriptional regulation in HEK293 cells

The predominant distribution of co-localized peaks around TSS implies that they may participate in the transcriptional regulation of their associated genes. RNA-seq analysis revealed that the RNA levels of co-localized G4s and R-loops-associated genes are significantly higher than all genes, G4s or R-loops- associated genes with the Mann-Whitney test p<2.2E-16 (*Figure 3A*). Different from G4s and R-loops, the co-localized G4s and R-loops are mainly localized within 1 kb of the TSS of transcriptionally active genes (~60% peaks with FPKM ≥ 5) (*Figure 3B*). To investigate the transcriptional regulation of co-localized G4s and R-loops in living cells, we performed the RNA-seq on HEK293 cells treated with and without ML216 or NSC617145 and then analyzed the differential gene expression using DESeq2 (*Love et al., 2014*). As a result, hundreds of genes were linked to co-localized G4s and R-loops with increased G4 signals (at least 1.5 foldchange) and at the same time exhibited significant changes in expression levels upon inhibition of BLM or WRN in HEK293 cells (*Figure 3C*), suggesting that co-localized G4s and R-loops could regulate the transcription of their associated genes. Distribution analysis showed that these differential co-localized G4s and R-loops are mainly localized in the promoter-TSS (*Figure 3D*). Among the differential genes, 125 genes were co-regulated by both ML216 and NSC617145 (*Figure 3E*), suggesting that BLM and WRN could co-regulate the transcription of genes by resolving G4s. Gene ontology (GO) analysis showed that co-localized G4s & R-loops-regulated genes in HEK293 cells are mainly involved in cell cycle regulation, DNA/mRNA metabolic regulation, DNA damage response, chromatin binding, kinase binding, cell-substrate junction, et al (*Figure 3F*).

## Mapping of native co-localized G4s and R-loops in mESCs

mESCs are pluripotent stem cells that c differentiate into various types of cells of three germ lineages (*Murry and Keller, 2008*; *Young, 2011*). To understand the regulatory roles of co-localized G4s and R-loops in mESCs, we performed the HepG4-seq and HBD-seq on mESCs and finally uncovered 68,482 native overlapping peaks in mESCs, ranging in size from 100 bp to ~2 kb (*Figure 4A–D*, *Figure 4—figure supplement 1*, *Supplementary file 2*). Notably, unlike HEK293 cells, a large number of native G4s (95,128) were identified by HepG4-seq in mESCs (*Figure 4A*, *Supplementary file 2*), which well overlap with the PQS predicted by pqsfinder (*Figure 4B*), suggesting that native G4s exhibit obvious cell type-specific distribution.

For the genomic distribution of co-localized G4s and R-loops, unlike HEK293 cells, only 34.2% peaks are localized in promoters, exon1, and intron1 while 25.9% peaks in intergenic regions (*Figure 4E*). The distinct number and localization feature of co-localized G4s and R-loops in HEK293 cells and mESCs shows the cell type-specific distribution. The metagene analysis of overlapping peaks exhibited a distinct peak around TSS (*Figure 4F*), suggesting the potential for transcriptional regulation. The motifs enrichment analysis found that G-rich sequences are highly enriched in these overlapping peaks, similar to those in HEK293 cells (*Figure 4G*). Representative peaks were found in several key regulatory genes of mESCs (*Figure 4H*).

## Characterization of native co-localized G4s and R-loops in mESCs

Similar to HEK293 cells, the RNA levels of co-localized G4s and R-loops-associated genes were seen to be significantly higher in mESCs (Mann-Whitney test p<2.2E-16) (*Figure 5A*). However, unlike in HEK293 cells (*Figure 3B*), the overlapping peaks in mESCs are mainly localized in the proximal promoters (1 kb from TSS, 9063 peaks with FPKM ≥ 5) and the region 5–50 kb from the TSS of transcriptionally active genes (32,690 peaks with FPKM ≥ 5) (*Figure 5B*), suggesting that co-localized G4s and R-loops are possibly distributed in active promoters or enhancers. To test this idea, we analyzed the co-localization of G4 (HepG4-seq), R-loop (HBD-seq), ChIP-seq signals of multiple chromatin

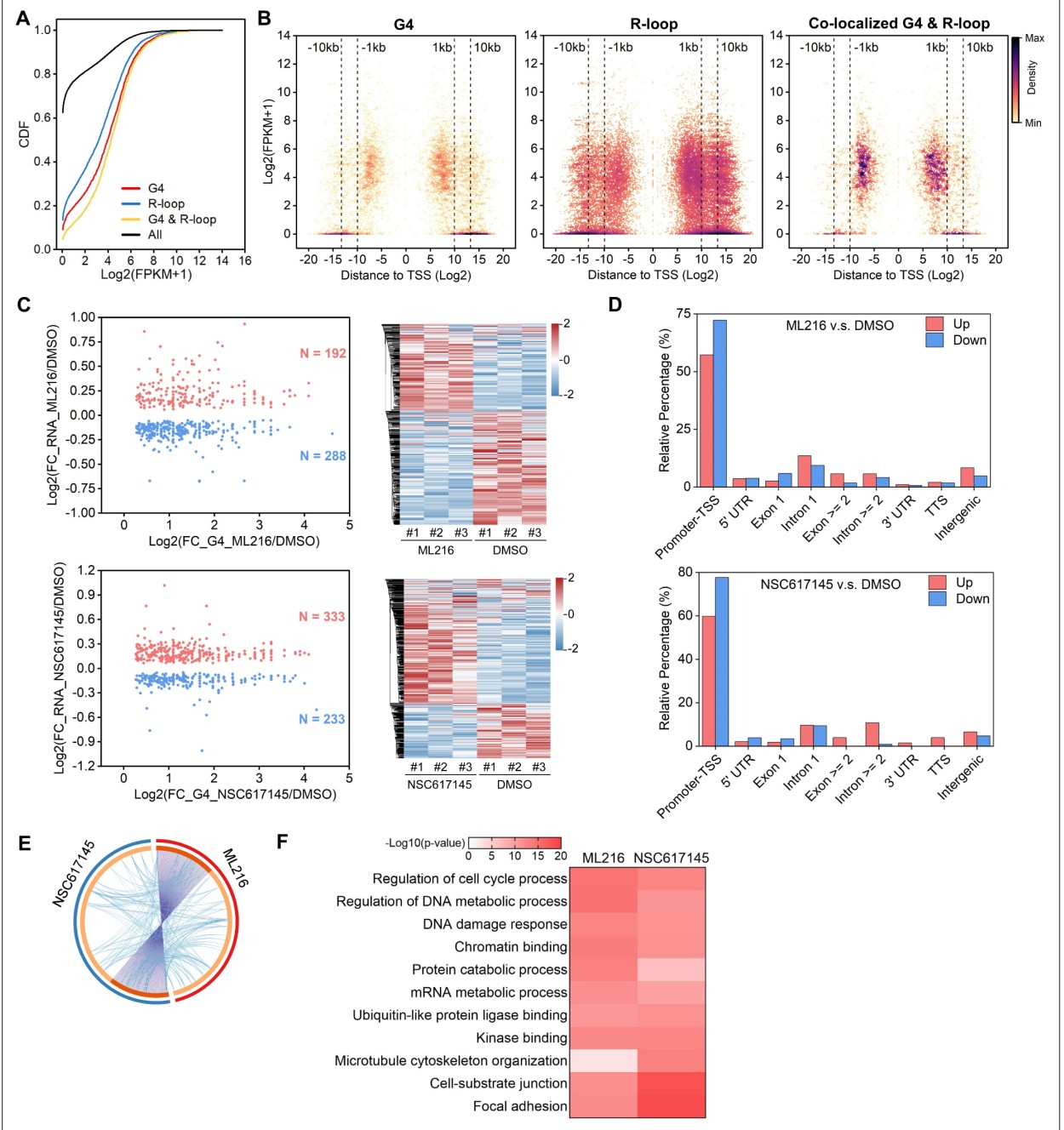

**Figure 3.** The co-localized G-quadruplexes (G4s) and R-loops-mediated transcriptional regulation in HEK293 cells. (**A**) Cumulative distribution plot showing comparisons of FPKMs of G-quadruplexe (G4), R-loop, co-localized G4 & R-loop associated genes and all genes in HEK293 cells. (**B**) Scatter plot showing the distributions of FPKMs of G4, R-loop, co-localized G4 & R-loop-associated genes versus the distances of G4, R-loop, co-localized G4 & R-loop to nearest transcription start site (TSS). The distance is in bp. Color scales represent the density of dots. (**C**) Left: Scatter plot showing the distributions of foldchanges (FCs) of RNA levels of co-localized G4 & R-loop -associated genes (p-value <0.05) versus FCs of G4 signals of co-localized G4s & R-loops (FC ≥ 1.5) after treatment with indicated inhibitors of G4 resolving helicases Bloom syndrome protein (BLM) or Werner syndrome ATP-dependent helicase (WRN). The number of genes were labeled on the plot. Right: Heatmap showing differential expression levels of co-localized G4 & R-loop-associated genes after treatment with indicated G4 inhibitors. RNA-seq data are from three biologically independent repeats. Color scales represent the normalized expression levels. (**D**) Distributions of co-localized G4s & R-loops expressions of which-associated genes were significantly up- or down-regulated after treatment with indicated G4 inhibitors across different gene features. (**E**) Circos plot showing the overlap co-localized G4 & R-loop-associated genes differentially expressed after the treatment of ML216 or NSC617145 in HEK293 cells. Purple lines link the same gene that ares shared by multiple groups. Blue lines link the genes, although different, fall under the same ontology term. Dark orange color of the inside arc represents the genes that are shared by multiple groups and the light orange color of the inside arc represents genes that are unique to that group. (**F**)

*Figure 3 continued on next page*

Figure 3 continued

Heatmap showing the GO-based enrichment terms of co-localized G4 & R-loop-associated genes differentially expressed after the treatment of ML216 or NSC617145 in HEK293 cells. The heatmap cells are colored by their p-values.

markers and RNA polymerase II with the phosphorylated serine 5 at its CTD domain (RNAP) that marks the transcriptionally initiated RNA Polymerase II (*Hsin and Manley, 2012*). As a result, co-localized G4s and R-loops were observed to well overlap with active chromatin markers (H3K4me3, H3K27ac, H3K36me3, H3K4me1) and RNAP but not the repressed chromatin marker H3K27me3 (*Figure 5C*).

Extensive studies define promoters into active, bivalent, and repressed states based on patterns of H3K4me3 and H3K27me3 (*Figure 5D*); enhancers are defined as active, poised, and unmarked states based on patterns of H3K27ac and H3K4me1 (*Figure 5D*; *Atlasi and Stunnenberg, 2017*; *Bernstein et al., 2006*; *Bibikova et al., 2008*; *Calo and Wysocka, 2013*; *Heintzman et al., 2007*). Notably, 20,741 and 19,726 co-localized G4s and R-loops are found in promoters and enhancers, respectively; 18,496 peaks are seen in active promoters; 15,787 peaks are seen in active enhancers (*Figure 5D*, *Supplementary file 2*). The co-localized G4s and R-loops in active promoters show high and almost equal signals of G4s and R-loops, and enrich H3K4me3, H3K27ac, and RNAP (*Figure 5E*). The co-localization of G4s, R-loops and RNAP at active promoters suggests that co-localized G4s and R-loops are likely linked to promoter-associated nascent RNAs (*Core et al., 2008*; *Li and Fu, 2019*; *Preker et al., 2008*; *Seila et al., 2008*). Interestingly, a medium level of H3K4me1 is present in a bimodal pattern beside co-localized G4s and R-loops at active promoters (*Figure 5E*). The co-localized G4s and R-loops in bivalent promoters exhibit low signals of both G4s and R-loops, and overlap with a high level of H3K27me3, a medium level of H3K4me1, and a low level of H3K4m3 (*Figure 5E*). The co-localized G4s and R-loops in active enhancers exhibit sharp peaks and well overlap with H3K27ac, H3K4me1, H3K4me3, and RNAP; the co-localized G4s and R-loops in poised enhancers enrich the active histone marks H3K4me1 and H3K4me3, and the repressive mark H3K27me3 while the signal of RNAP is very low (*Figure 5F*). The unmarked enhancers-associated co-localized G4s and R-loops show low levels of all marks tested. Given that enhancer RNAs (eRNAs) have been widely identified as non-coding RNAs in enhancers and are functionally important for enhancer activity (*Andersson et al., 2014*; *Kim et al., 2010*; *Sigova et al., 2015*), the co-occupancy of eRNAs are likely involved in the formation of co-localized G4s and R-loops in enhancers and shed light on the new regulatory mechanism of eRNA action.

## Modulation of co-localized G4s and R-loops by the helicase Dhx9

Dhx9 (also known as RNA Helicase A) is a versatile helicase capable of directly resolving R-loops and G4s or promoting R-loop formation by unwinding secondary structures in the nascent RNA strand (*Chakraborty and Grosse, 2011*; *Chakraborty et al., 2018*; *Cristini et al., 2018*; *Matsui et al., 2020*; *Tang et al., 2022*; *Yuan et al., 2021*), and has been reported to play roles in DNA replication, transcription, translation, RNA processing and transport and maintenance of genomic stability (*Aktaş et al., 2017*; *Aratani et al., 2001*; *Chellini et al., 2022*; *Jain et al., 2013*; *Tang et al., 2022*). Thus, Dhx9 is a promising regulator of co-localized G4s and R-loops.

To investigate the role of Dhx9 in modulating co-localized G4s and R-loops, we generated the Dhx9 knockout mESCs (*dhx9^{KO}*) by CRISPR/Cas9-mediated gene editing (*Ran et al., 2013*). The depletion of Dhx9 in the *dhx9^{KO}* mESC clone was confirmed by western blot assay (*Figure 6A*) and immunofluorescence staining (*Figure 6B*). We next examined the G4 an R-loop levels in the *dhx9^{KO}* mESCs by performing HepG4-seq and HBD-seq. Notably, compared to the wildtype mESCs, a large amount of G4s or R-loops within co-localized G4s and R-loops exhibited significantly up-regulated or down-regulated signals in the *dhx9^{KO}* mESCs (*Figure 6C*, *Figure 6—figure supplement 1A*, *Supplementary file 2*), suggesting that Dhx9 could unwind or promote the formation of co-localized G4s and R-loops in mESCs. Interestingly, only a small proportion of co-localized G4s and R-loops displayed differential G4s and R-loops at the same time in the *dhx9^{KO}* mESCs (*Figure 6D*, *Figure 6—figure supplement 1B*), suggesting that Dhx9 cannot simultaneously unwind or promote G4s and R-loops within co-localized G4 and R-loop regions and that multiple helicases or regulators are required for modulating these regions.

Given that co-localized G4s and R-loops have been shown to be enriched in active promoters and enhancers (*Figure 5D–F*), the loss-of-Dhx9-induced differential co-localized G4s and R-loops

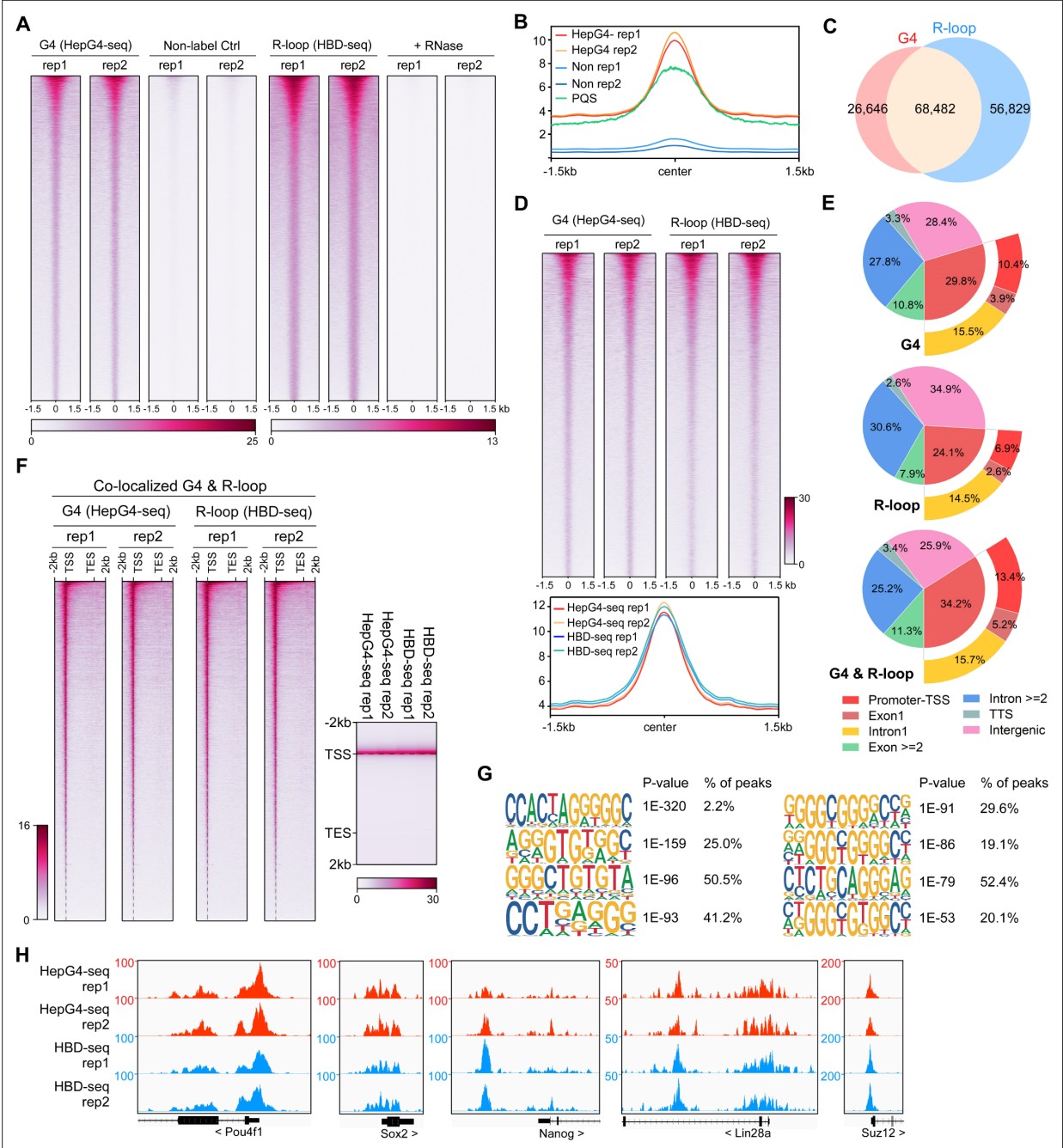

**Figure 4.** Mapping of the co-localized G-quadruplexes (G4s) and R-loops in the mouse embryonic stem cells. (**A**) Heatmap showing the signal of HepG4-seq and hybrid-binding domain (HBD)-seq ±1.5 kb around the center of peaks in mouse embryonic stem cells (mESCs). Two biologically independent replicates are shown. Two biologically independent non-label replicates were the negative controls for HepG4-seq. Two biologically independent replicates with the treatment of RNase A and RNase H prior to HBD-seq were the negative controls for HBD-seq. Color scales represent the density of the signals. (**B**) Profile plot showing the average signal of HepG4-seq reads ±1.5 kb around the center of peaks and the average max Scores of PQS calculated by pqsfinder at the same positions. HepG4 rep1/rep2, two biologically independent HepG4-seq replicates in mESCs; Non rep1/rep2, two biologically independent non-label negative control replicates in mESCs. (**C**) Venn diagram comparing the DNA G4 and R-loop in mESCs. (**D**) Top: Heatmap showing the signal of HepG4-seq and HBD-seq ±1.5 kb around the center of co-localized G4s & R-loops in mESCs. Two biologically independent replicates are shown. Color scales represent the density of the signals. Bottom: Profile plot showing the average signal of HepG4-seq and HBD-seq reads ±1.5 kb around the center of co-localized G4s & R-loops in mESCs. (**E**) Distribution of G4s, R-loops, and co-localized G4s & R-loops signals in mESCs in different gene features. (**F**) Left: Heatmap showing the signal of HepG4-seq and HBD-seq reads of the co-localized G4s & R-loops in mESCs along the gene body, 2 kb upstream of TSS and 2 kb downstream of TES. Right: Profile plot showing the average signal of HepG4-seq and HBD-seq reads of the co-localized G4s & R-loops in mESCs along the indicated gene features. The plot is visualized using the heatmap.

*Figure 4 continued on next page*

Figure 4 continued

Color scales represent the density of the signals. (**G**) The top enriched motifs of the co-localized G4s & R-loops in mESCs. (**H**) Representative genome browser tracks showing the HepG4-seq and HBD-seq signals of the co-localized G4s & R-loops in mESCs along the indicated genomic loci.

The online version of this article includes the following figure supplement(s) for figure 4:

**Figure supplement 1.** Distribution of co-localized G4 & R-loop peak sizes.

preferentially localize in the active and bivalent promoters and all three types of enhancers (*Figure 6E*). To explore the effect of Dhx9 on the transcription of co-localized G4s and R-loops -associated genes, we performed the RNA-seq on wild-type and *dhx9$^{KO}$* mESCs. Differential gene expression analysis revealed 1647 significantly up-regulated genes and 1916 significantly down-regulated genes in the absence of Dhx9 (*Figure 6—figure supplement 1C, D*). Importantly, loss of Dhx9 resulted in hundreds of G4-, R-loop-, and G4&R-loop-associated genes with significantly differential expression (*Figure 6F*), suggesting that Dhx9 could regulate transcription by modulating G4s, R-loops, and co-localized G4s and R-loops. Representative Dhx9-regulated locus are shown in *Figure 6G*. GO analysis showed that co-localized G4s and R-loops-associated genes that showed differential expression after knocking out Dhx9 are mainly involved in negative regulation of cell differentiation, head development, positive regulation of cell motility, DNA-binding transcription activator activity, pattern specification process, embryonic organ morphogenesis, et al., suggesting that Dhx9 may regulate the cell fate of mESCs by modulating co-localized G4s and R-loops (*Figure 6H*). Coinciding with GO analysis, Dhx9 knockout in the mouse causes embryonic lethality, and Dhx9 knockdown leads to large structural changes in chromatin and eventually cell death (*He et al., 2008*; *Zhang et al., 2004*). Heterozygous loss-of-function variants of DHX9 are associated with neurodevelopmental disorders in humans (*Yamada et al., 2023*).

## Characterization of co-localized G4s and R-loops directly bound by Dhx9

Tens of helicases or regulators have been reported to directly resolve or stabilize G4s or R-loops (*Mendoza et al., 2016*; *Varshney et al., 2020*; *Yang et al., 2023*). Interestingly, the loss of Dhx9 caused 30 of these helicases/regulators to be significantly differentially expressed (*Figure 7A*) and Dhx9 physically interacts with at least 10 of them based on the STRING protein-protein interaction network database (*Szklarczyk et al., 2023*; *Figure 6—figure supplement 1E*). These data suggest that Dhx9 could also indirectly modulate G4s and R-loops by affecting other helicases or regulators. Thus, to explore the direct target co-localized G4s and R-loops of Dhx9, we performed the CUT&Tag-seq using the Dhx9 antibody (*Kaya-Okur et al., 2019*) and revealed 54,982 Dhx9 binding peaks in wild-type mESCs (*Figure 7B*). Notably, 65.5% Dhx9 binding peaks well overlapped with 69.9% co-localized G4s and R-loops in mESCs (*Figure 7C and D*), suggesting that Dhx9 is a direct and major regulator of co-localized G4s and R-loops in mESCs. Motif analysis showed that G-rich sequences are highly enriched in the Dhx9 binding peaks overlapping with co-localized G4s and R-loops in mESCs (*Figure 7E*), further demonstrating that Dhx9 directly bind to co-localized G4s and R-loops that harbor G-rich sequences as shown in *Figure 4G*.

We next compared the Dhx9-bound co-localized G4s and R-loops in wild-type and *dhx9$^{KO}$* mESCs, and identified 1382 significantly up-regulated peaks (823 with increased G4s and 559 with increased R-loops) and 4789 significantly down-regulated peaks (2278 with decreased G4s and 2511 with decreased R-loops) (*Figure 7F*, *Figure 6—figure supplement 1F*), accounting for ~50–75% of differential co-localized G4s and R-loops in absence of Dhx9 (*Figure 6C*). Analysis of the genomic distribution of Dhx9-bound differential co-localized G4s and R-loops found that these peaks are mainly localized in the active and bivalent promoters and all three types of enhancers (*Figure 7G*). The Dhx9-bound differential co-localized G4s and R-loops were linked to 852 genes with significantly differential expression (*Figure 6—figure supplement 1G*, *Supplementary file 2*), which are enriched in GO terms related to pattern specification, cell junction organization, brain development, negative regulation of cell differentiation, mesenchyme/mesoderm development, embryonic morphogenesis, et al. (*Figure 7H*). Several key regulators of mouse embryonic stem cell and embryonic development, such as Nanog, Lin28a, Bmp4, Wnt8a, Gata2, and Lef1, were shown to be transcriptionally regulated by Dhx9 through direct modulation of their associated G4s and R-loops (*Figure 7I*). These

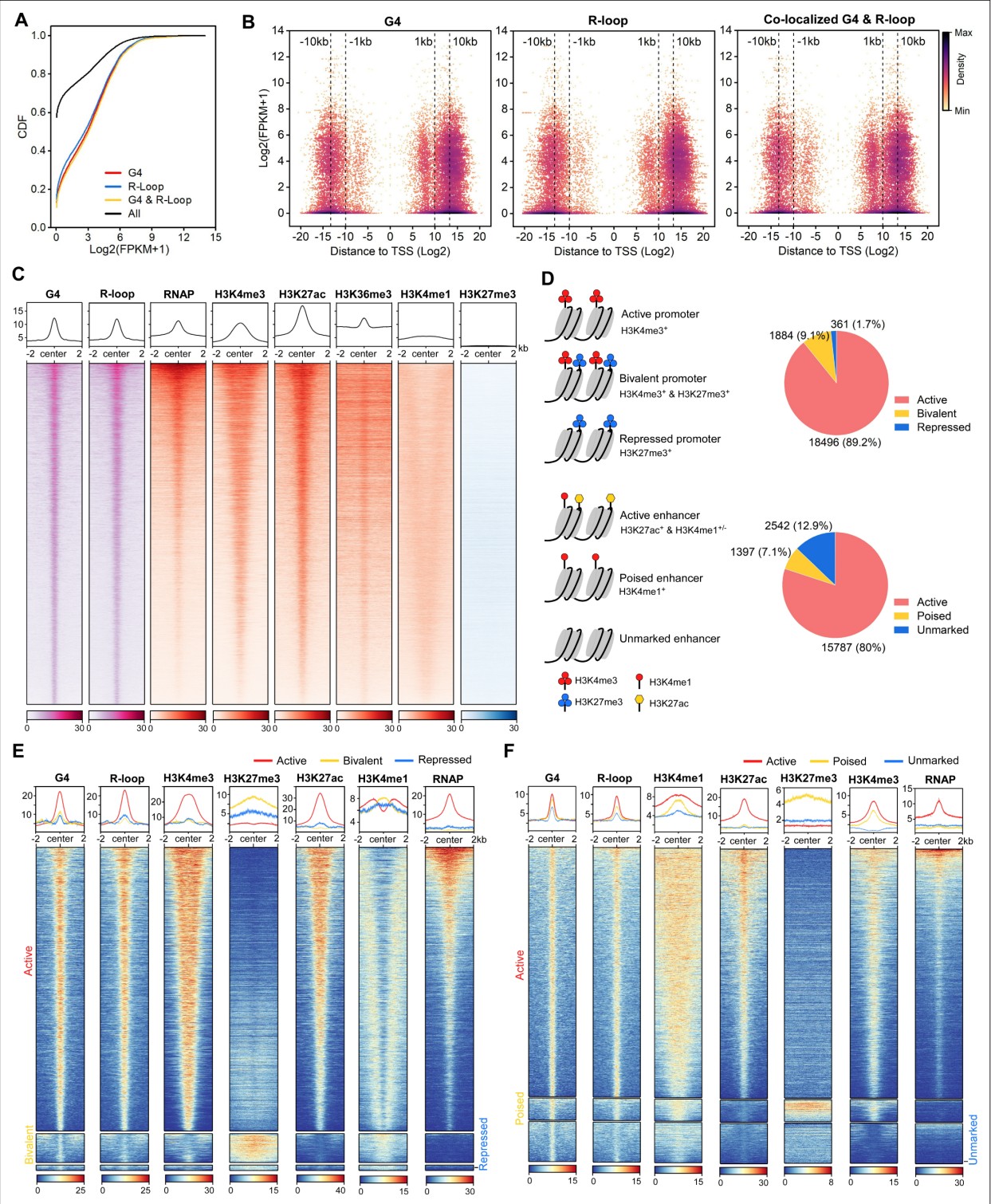

**Figure 5.** The co-localized G-quadruplexes (G4s) & R-loops are mainly localized in active promoters and enhancers. (**A**) Cumulative distribution plot showing comparisons of FPKMs of G4-, R-loop-, and co-localized G4 & R-loop-associated genes and all genes in mouse embryonic stem cells (mESCs). (**B**) Scatter plot showing the distributions of FPKMs of G4-, R-loop-, and co-localized G4 & R-loop -associated genes versus the distances of these peaks to the nearest transcription start site (TSS). The distance is in bp. Color scales represent the density of dots. (**C**) Heatmap showing the signal of representative HepG4-seq (G4), HBD-seq (R-loop), RNA polymerase II Ser5P, H3K4me3, H3K27ac, H3K36me3, H3K4me1, and H3K27me3±2 kb around the center of the co-localized G4s & R-loops in mESCs. Color scales represent the density of the signals. The average signal is plotted at the top of each heatmap panel. (**D**) Schematic of different types of promoters and enhancers. Pie chart showing the proportion of different types of promoters

*Figure 5 continued on next page*

*Figure 5 continued*

or enhancers that harbor the co-localized G4s & R-loops. (**E, F**) Heatmap showing the signal of representative HepG4-seq (G4), HBD-seq (R-loop), RNA polymerase II Ser5P, H3K4me3, H3K27ac, H3K36me3, H3K4me1, and H3K27me3±2 kb around the center of the co-localized G4s & R-loops in the different types of promoters (**E**) or enhancers (**F**) in mESCs. Color scales represent the density of the signals. The average signal is plotted at the top of each heatmap panel.

data suggest that Dhx9 significantly contributes to transcriptional regulation of co-localized G4s and R-loops-associated genes.

## Dhx9 regulates the cell fate of mESCs

To understand the role of Dhx9 in regulating the cell fate of mESCs, we first examined the RNA levels of several key genes that maintain the pluripotency of mESCs by quantitative RT-PCR (qRT-PCR) and found that the RNA levels of Lin28a and Oct4 were significantly decreased and the RNA level of Nanog was significantly increased when Dhx9 was knocked out (**Figure 8A**). Western blot assay showed that *dhx9*^KO^ mESCs produce obviously lower level of Lin28a protein than wild-type mESCs, consistent with its RNA level, whereas in contrast to the RNA level, the protein level of Nanog was significantly decreased in *dhx9*^KO^ mESCs, suggesting that Dhx9 directly or indirectly modulates the translation of Nanog (**Figure 8B**). In line with the western blot assay, immunofluorescence staining of WT and *dhx9*^KO^ mESCs showed that the loss of Dhx9 leads to reduced protein level of Nanog, but not Oct4, while *dhx9*^KO^ mESCs exhibited normal morphology (**Figure 8C**). The mESCs can be maintained in a proliferative state for prolonged periods, which was known as 'self-renewal' (**Liang and Zhang, 2013**; **Murry and Keller, 2008**). Nanog and Lin28a have been reported to promote embryonic stem self-renewal (**Chambers et al., 2003**; **Mitsui et al., 2003**; **Xu et al., 2009**). Coinciding with reduced levels of Nanog and Lin28a proteins, *dhx9*^KO^ mESCs was shown to modestly arrest at the S phase of the cell cycle (**Figure 8D**) and exhibited significantly attenuated proliferation capacity (**Figure 8E**), suggesting that Dhx9 regulates the self-renewal of mESCs.

mESCs are pluripotent stem cells which are able to differentiate into three germ lineages (**Murry and Keller, 2008**; **Young, 2011**). In the absence of differentiation inhibitor LIF, mESCs cultured in suspension spontaneously form three-dimensional aggregates called embryoid bodies (EBs), which could recapitulate many aspects of early embryogenesis, including the induction of three early germ lineages (**Simunovic and Brivanlou, 2017**). To understand the role of Dhx9 in regulating the pluripotency of mESCs, we performed the EB assay using the wild-type and *dhx9*^KO^ mESCs. As the EB differentiation progressed, loss of Dhx9 resulted in apparently smaller and fewer EBs than wild-type cells (**Figure 8F**). At the same time, we collected EBs at different days of EB differentiation and examined the RNA levels of well-known markers of three germ lineages by qRT-PCR. As shown in **Figure 8G**, all maker genes tested displayed significantly differential expression, suggesting that Dhx9 regulates the pluripotency of mESCs, which is in line with the GO enrichment results of Dhx9-regulated co-localized G4s and R-loops-associated genes in **Figure 7H**. Taken together, Dhx9 regulates the self-renewal and differentiation capacities of mESCs.

## Comparisons of HepG4-seq and HBD-seq with previous methods

To compare the performance of HepG4-seq and BG4 CUT&Tag, we analyzed the data using the same bioinformatic pipeline. As shown in **Figure 9A**, 80% (5,459) and 71% (67,935) HepG4-seq peaks overlap with BG4 CUT&Tag peaks in HEK293 cells and mESCs, respectively. SEACR is a highly selective peak caller for CUT&Tag and CUT&Run (**Meers et al., 2019**). With the help of SEACR, BG4, CUT&Tag obtained a much larger number of peaks in HEK293 cells and mESCs than previously reported MACS2-identified peaks. Surprisingly, 70,954 and 103,303 consensus peaks were identified by BG4 CUT&Tag alone in HEK293 cells and mESCs, respectively (**Figure 9A**), suggesting that BG4 antibody may promote the folding of G4 sequences during BG4-seq or that BG4 antibody is more sensitive than G4-hemin-induced proximal biotinylation in recognizing native G4s. As shown in **Figure 9B**, the patterns of BG4 CUT&Tag peaks overlap well with those of PQS while HepG4-seq peaks exhibit a much sharper shape and better resolution around the center of PQS patterns, indicating that BG4 may promote the folding of G4 motifs. Interestingly, the signal intensities of the HepG4-seq peaks are significantly lower than those of BG4 CUT&Tag in HEK293 cells, but obviously higher than those of BG4 CUT&Tag in mESCs. In the in vitro hemin-G4-induced self-biotinylation assay, parallel G4s exhibit

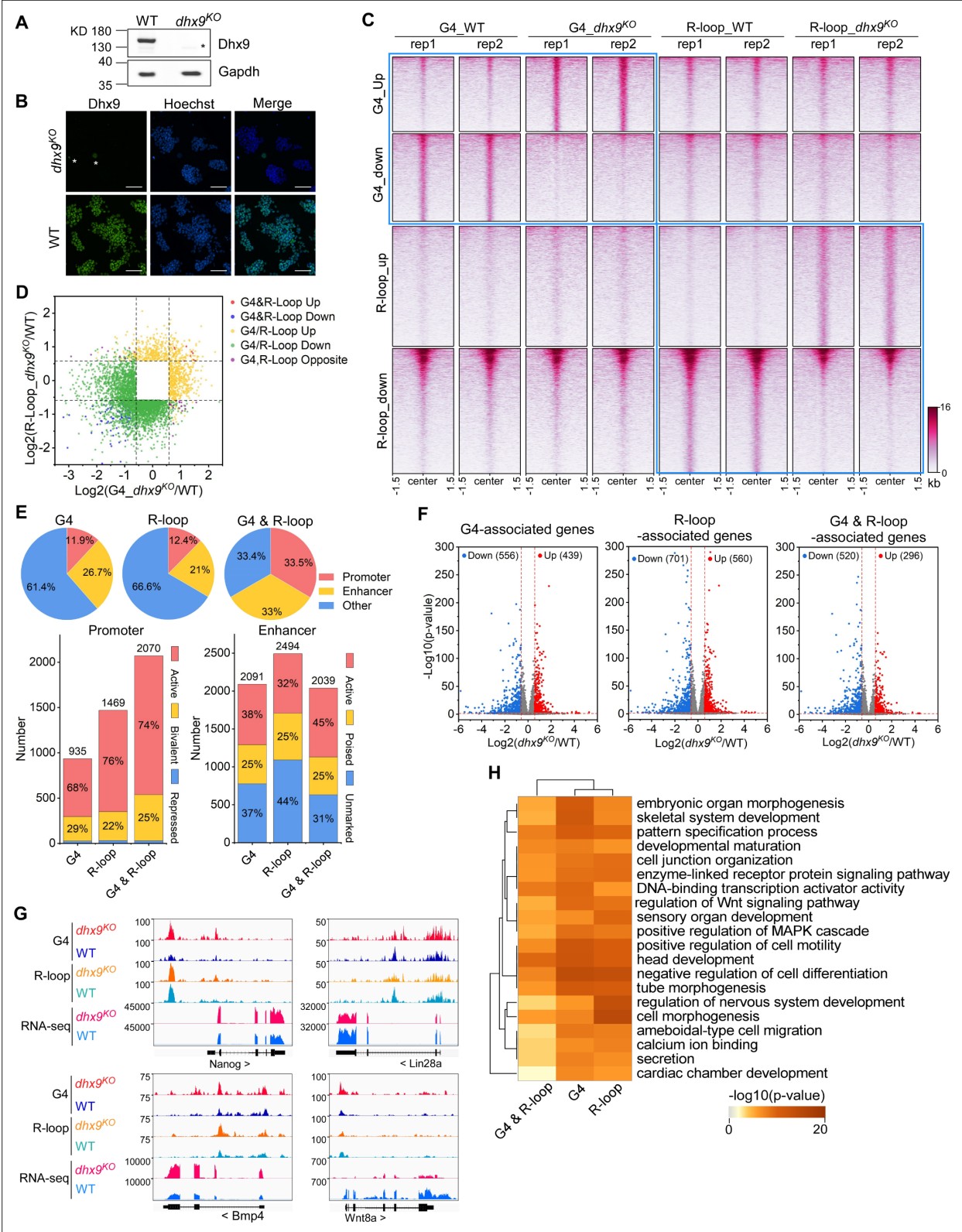

**Figure 6.** Modulation of the co-localized G-quadruplexes (G4s) & R-loops by the helicase Dhx9. (**A**) Western blot showing the protein levels of Dhx9 and Gapdh in the wild-type (WT) and *dhx9^KO* mouse embryonic stem cells (mESCs). The non-specific band is labeled with a star. This experiment was repeated once with similar results. (**B**) Immunofluorescence staining of Dhx9 in the WT and *dhx9^KO* mESCs cultured without mouse embryonic fibroblasts (MEFs) feeder. Nuclei were stained with the Hoechst33342. The immunofluorescence signal of contaminated MEFs is labeled with a a star. Scale bar,

*Figure 6 continued on next page*

*Figure 6 continued*

100 µm. (**C**) Heatmap showing the signal of DNA G4 (HepG4-seq) and R-loop (HBD-seq) ±1.5 kb around the center of significantly differential peaks (p-value <0.05, fold change ≥1.5) in WT and *dhx9*[KO] mESCs. Two biologically independent replicates are shown. Color scales represent the density of the signals. (**D**) Scatter plot showing distributions of foldchanges of differential G4s versus foldchanges of differential R-loop in WT and *dhx9*[KO] mESCs. 'G4&R-loop Up,' both G4 and R-loop up-regulated; 'G4&R-loop Down,' both G4 and R-loop down-regulated; 'G4/R-loop Up,' G4 or R-loop up-regulated; 'G4/R-loop Down,' G4 or R-loop down-regulated; 'G4, R-loop Opposite,' G4 up-regulated and R-loop down-regulated, or, G4 down-regulated and R-loop up-regulated. (**E**) Top: pie chart showing proportions of differential G4s, R-loops, and co-localized G4s & R-loops in *dhx9*[KO] mESCs in promoters, enhancers, and other regions; Bottom: bar chart showing numbers of differential G4, R-loops and co-localized G4s & R-loops in *dhx9*[KO] mESCs in different types of promoters or enhancers. (**F**) Volcano plot showing distributions of G4-, R-loop-, or co-localized G4 & R-loop-associated genes differentially expressed in WT and *dhx9*[KO] mESCs. Significantly up-regulated (p-value <0.05, fold change ≥1.5) and down-regulated (p-value <0.05, fold change ≤0.67) genes in the *dhx9*[KO] mESCs are labeled with red and blue dots, respectively. The numbers of up- or down-regulated genes are labeled on the plot. (**G**) Representative genome browser tracks showing the G4 (HepG4), R-loop (HBD-seq), and RNA-seq signals in WT and *dhx9*[KO] mESCs along the indicated genomic loci. (**H**) GO-based enrichment terms of G4-, R-loop-, or co-localized G4 & R-loop-associated genes differentially expressed in WT and *dhx9*[KO] mESCs were hierarchically clustered into a tree based on Kappa-statistical similarities among their gene memberships. The heatmap cells are colored by their p-values.

The online version of this article includes the following source data and figure supplement(s) for figure 6:

**Source data 1.** PDF file containing original western blots for *Figure 6A* and original microscope images for *Figure 6B*, indicating the selected regions.

**Source data 2.** Original files for western blots are displayed in *Figure 6A* and microscope images are displayed in *Figure 6B*.

**Figure supplement 1.** This figure is linked to both *Figure 6* and *Figure 7*.

higher peroxidase activities than anti-parallel G4s (*Lat et al., 2020*). The signal intensity changes of HepG4-seq in different cells possibly reflect the dynamics of G4 conformation. In the future, people may need to combine HepG4-seq and BG4s-eq to carefully explain the endogenous G4s.

During the past decade, R-loops were mapped using either S9.6 monoclonal antibody or catalytically inactive ribonuclease H1 for specific DNA-RNA hybrids. First, we compared the HBD-seq with S9.6 CUT&Tag in mESCs using the same bioinformatic pipeline and SEACR for peak calling. As a result, HBD-seq and S9.6 CUT&Tag identified similar numbers of peaks (125,311 for HBD-seq, 154,993 for S9.6 CUT&Tag), and 45,989 HBD-seq peaks overlap with 88,036 S9.6 peaks, indicating that S9.6 peaks are possibly narrower than HBD-seq peaks. As shown in *Figure 9D*, the overlapping peaks showed similar heatmap patterns and signal intensities between HBD-seq and S9.6 CUT&Tag while, notably, peaks of S9.6 CUT&Tag exhibit a much sharper signals at the center of peaks, which reflect the difference of binding modules of HBD and S9.6. GST-2xHBD and S9.6 mediated CUT&Tag were shown to generate highly similar native R-loop profiles (*Wang et al., 2021*). HBD-seq, as an optimized version of GST-2xHBD CUT&Tag, still generated ~80,000 extra peaks compared to S9.6 CUT&Tag, suggesting that the status of the recombinant 2xHBD protein could substantially affect R-loop mapping. To understand the difference among inactive ribonuclease H1-based methods, we compared the HBD-seq, MapR (inactive ribonuclease H1-mediated CUT&Run) (*Yan et al., 2019*) and GST-2xHBD CUT&Tag (*Wang et al., 2021*) and re-analyzed the data using the same pipeline and peak caller SEACR. In HEK293 cells, HBD-seq, MapR, and GST-2xHBD CUT&Tag identified 42,488, 12,520, and 17,570 peaks, respectively; 5752 MaR peaks and 11,701 GST-2xHBD CUT&Tag peaks overlap with HBD-seq peaks (*Figure 9E*). Notably, the heatmap of all HBD-seq peaks showed an obviously stronger signal intensities and sharper peak shapes than MapR and GST-2xHBD CUT&Tag, suggesting that HBD-seq provides superior R-loop signals. In consideration of the difference in DNA-RNA hybrid binding affinity and/or specificity between HBD and S9.6, the HBD-seq and S9.6 CUT&Tag may need to be combined together to carefully explain the native R-loops.

## Discussion

In this study, we developed the new method 'HepG4-seq' and optimized the RNase H1 HBD domain-based HBD-seq to robustly map endogenous G4s and R-loops, respectively, in living cells with high specificity. Using the HepG4-seq and HBD-seq, we systematically characterized the native co-localized G4s and R-loops in HEK293 cells and mESCs, and revealed that co-localized G4s and R-loops are dynamically altered in a cell type-dependent way and largely localized at active promoters and enhancers of transcriptional active genes. Small molecules-induced inhibition of helicases BLM or WRN resulted in a significant accumulation of G4s within co-localized G4s and R-loops and at the

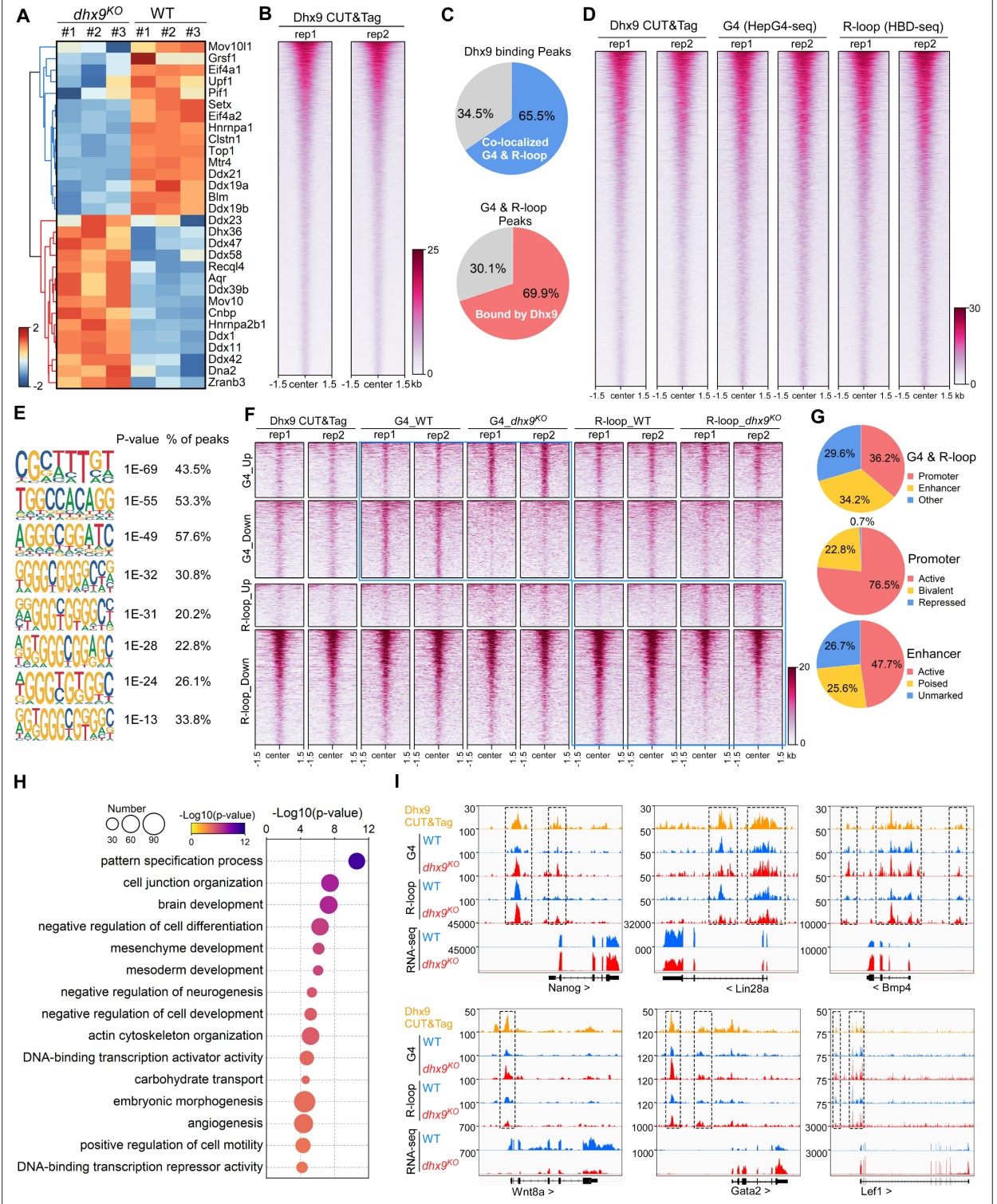

**Figure 7.** Characterization of the co-localized G-quadruplexes (G4s) & R-loops directly bound by Dhx9. (**A**) Heatmap showing expression levels of resolving helicases or regulators of G4s and/or R-loops differentially expressed in wild-type (WT) and *dhx9*^KO mouse embryonic stem cells (mESCs). Color scales represent the normalized expression levels. (**B**) Heatmap showing the signal of Dhx9 CUT&Tag reads ±1.5 kb around the center of peaks in WT mESCs. Two biologically independent replicates are shown. Color scales represent the density of the signals. (**C**) Pie chart showing proportions of the overlapping peaks between Dhx9 binding peaks and the co-localized G4 & R-loops peaks. (**D**) Heatmap showing the signal of Dhx9 CUT&Tag, HepG4-seq, and HBD-seq ±1.5 kb around the center of Dhx9-bound Co-localized G4 & R-loop peaks in mESCs. Two biologically independent replicates are shown. Color scales represent the density of the signals. (**E**) The top enriched motifs of Dhx9 binding peaks overlapping with the co-localized G4 &

*Figure 7 continued on next page*

*Figure 7 continued*

R-loops in mESCs. (**F**) Heatmap showing the signal of Dhx9 CUT&Tag, HepG4-seq and HBD-seq ±1.5 kb around the center of Dhx9-bound significantly differential Co-localized G4 & R-loop peaks (p-value <0.05, fold change ≥1.5) in WT and *dhx9*^KO mESCs. Two biologically independent replicates are shown. Color scales represent the density of the signals. (**G**) Pie chart showing proportions of differential Dhx9-bound co-localized G4 & R-loops in *dhx9*^KO mESCs in different types of promoters or enhancers. (**H**) Top enriched GO terms in Dhx9-bound co-localized G4 & R-loop-associated genes that are differentially expressed in WT and *dhx9*^KO mESCs. The bubble size represents the number of genes in each indicated term. The color scale represents the p-value. (**I**) Representative genome browser tracks showing the Dhx9 CUT&Tag, G4 (HepG4), R-loop (HBD-seq), and RNA-seq signals in WT and *dhx9*^KO mESCs along the indicated genomic loci. The dashed box highlights Dhx9-bound significantly differential co-localized G4 & R-loop peaks in *dhx9*^KO mESCs.

same time led to genes with significantly differential expression in HEK293 cells that are enriched in the processes related to cell cycle, DNA metabolic, DNA damage response, chromatin binding, et al. Furthermore, we characterized the helicase Dhx9 as a key regulator of co-localized G4s and R-loops which efficiently unwinds or promotes co-localized G4s and R-loops, and illustrated that depletion of Dhx9 significantly altered the transcription of co-localized G4s and R-loops-associated genes that are enriched in embryonic development, cell differentiation, and germ lineage development, et al. Therefore, the loss of Dhx9 apparently impaired the self-renewal and pluripotency of mESCs.

In this study, we utilized a low dosage of hemin, similar to the physiological concentration in normal human erythrocytes, to spark the peroxidase activity of endogenous G4s without significantly altering the levels of native G4s (***Figure 1—figure supplement 1B, C***) and further robustly biotinylated G4s themselves by G4-hermin complex-mediated proximity labeling in just 1 min in living cells (***Cheng et al., 2009***; ***Einarson and Sen, 2017***; ***Lat et al., 2020***; ***Li et al., 2016***; ***Stadlbauer et al., 2021***; ***Yang et al., 2011***). In consideration of the high affinity and specificity, the recombinant streptavidin monomer (***Lim et al., 2013***) is able to recognize the biotinylated G4s with high sensitivity and specificity and thereby yield robust CUT&Tag signals with the help of Moon-tag system (***Boersma et al., 2019***). Therefore, our HepG4-seq strategy is able to robustly and specifically capture native G4s. In HEK293 cells, HepG4-seq uncovered 6799 consensus G4 peaks under wild-type status and 77,003 G4 peaks in the presence of BLM/WRN inhibitors, suggesting that HepG4-seq is capable of detecting endogenous G4s with high sensitivity. Yet, the hemin-G4 complex-induced peroxidase activities were reported to be variable between paralleled and anti-paralleled G4s in the in vitro assay (***Lat et al., 2020***), although only 2–4 synthesized short DNA fragments were used for the test. Given that a method of genome-widely distinguishing paralleled and anti-paralleled G4s is lacking, HepG4-seq and BG4-seq may be combined to characterize the conformation of G4s.

Notably, we also discovered that the native co-localized G4s and R-loops landscape is altered in a cell-dependent manner and that approximately 10 folds more peaks were observed in mESCs than in HEK293 cells, which reflects that co-localized G4s and R-loops have the potential to regulate the complex pluripotency network in mESCs. CTCF is a key regulator of genome organization and gene expression (***Ong and Corces, 2014***). Recently, Wulfridge et al. reported that CTCF-bound regions are enriched for both R-loops, and G4s and G4s associated with R-loops promote CTCF binding (***Wulfridge et al., 2023***). Interestingly, the enriched motif with the most significant p-value in mESCs co-localized G4s and R-loops (***Figure 4G***) well matches the motif of CTCF ChIP-seq, suggesting that co-localized G4s and R-loops may be able to modulate the CTCF binding. To be noticed, the co-localized G4s and R-loops were identified by bioinformatically intersection analysis. The heterogenous distribution between cells will give false positive co-occurrence patterns. Recently, multi-CUT&Tag, a method for simultaneous mapping of multiple chromatin proteins, has been reported (***Bartosovic and Castelo-Branco, 2023***; ***Gopalan et al., 2021***; ***Meers et al., 2023***). As the multi-CUT&Tag, we could combine the HepG4 and HBD-seq to simultaneously map the co-localized G4s and R-loops even in the single cell level.

Furthermore, while 47,857 co-localized G4s and R-loops are directly bound by Dhx9 in the wild-type mESCs (***Supplementary file 2***), only 4060 of them display significantly differential signals in the absence of Dhx9. In addition, a limited number of co-localized G4s and R-loops showed differential G4s and R-loops at the same time in the *dhx9*^KO mESCs. These data suggest that redundant regulators exist. It is worth noting that depletion of Dhx9 significantly altered the transcription of 30 known G4s and/or R-loops helicases/regulators (***Figure 7A***) and that half of these helicases/regulators are able to establish physical interaction network (***Figure 6—figure supplement 1E***). Due to the redundancy

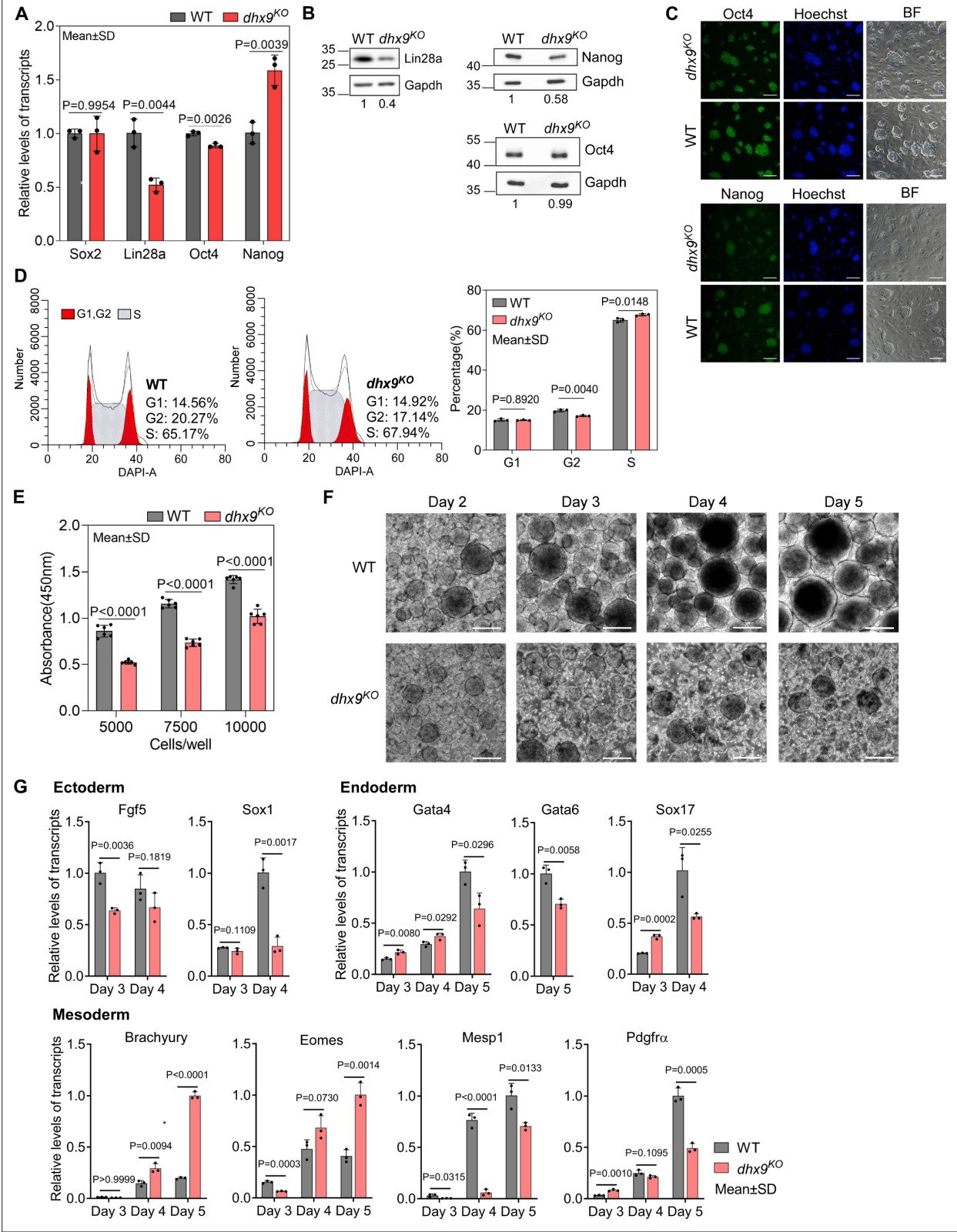

**Figure 8.** Dhx9 regulates the cell fate of mouse embryonic stem cells. (**A**) Relative RNA levels of indicated genes in the wild-type (WT) and *dhx9*$^{KO}$ mouse embryonic stem cells (mESCs) that were measured by quantitative RT-PCR (qRT-PCR). Data are means ± SD; n=3 (three independent biological samples), significance was determined using the two-tailed Student's *t*-test. (**B**) Western blot showing the protein levels of indicated genes in the WT and *dhx9*$^{KO}$ mESCs. The normalized relative protein levels were labeled below each panel, where gel images were quantified by ImageJ and the level

*Figure 8 continued on next page*

*Figure 8 continued*

of Gapdh was used for normalization. This experiment was repeated once with similar results. (**C**) Immunofluorescence staining of Oct4 and Nanog in the WT and *dhx9^KO^* mESCs cultured on MEF feeder. Nuclei were stained with the Hoechst33342. BF, bright field. Scale bar, 100 µm. (**D**) Cell cycle profiles determined by flow cytometry of DAPI staining in the WT and *dhx9^KO^* mESCs. The proportions of different phases of the cell cycle were analyzed by ModFit. Data are means ± SD; n=3, and significance was determined using the two-tailed Student's *t*-test. (**E**) Cell proliferation rate of WT and *dhx9^KO^* mESCs were measured by CCK-8 cell proliferation assay. The number of cells seeded at the beginning was labeled at the x-axis. Absorbance at 450 nm was determined after 2 days of culture. Data are means ± SD; n=6, and significance was determined using the two-tailed Student's *t*-test. (**F**) Pictures of embryoid bodies at indicated days of in vitro differentiation of WT and *dhx9^KO^* mESCs. Scale bar, 200 µm. (**G**) Relative RNA levels of indicated genes in the WT and *dhx9^KO^* embryoid bodies at indicated days of in vitro differentiation that were measured by qRT-PCR. Data are means ± SD; n=3, and significance was determined using the two-tailed Student's *t*-test.

The online version of this article includes the following source data for figure 8:

**Source data 1.** PDF file containing original western blots for *Figure 8B* and original microscope images for *Figure 8C and F*, indicating the selected regions.

**Source data 2.** Original files for western blots are displayed in *Figure 8B* and microscope images are displayed in *Figure 8C and F*.

and compensatory roles of G4 and R-loop regulators, a comprehensively mechanistic study of G4 and R-loop interplay in living cells is difficult and lacking in the field. A degron system-mediated simultaneous and/or stepwise degradation system of multiple regulators will help us elucidate the interplaying effects between G4s and R-loops.

Taken together, our study provides new insights into exploring the regulatory roles of co-localized G4s and R-loops in development and disease.

# Materials and methods

## Key resources table

| Reagent type (species) or resource | Designation | Source or reference | Identifiers | Additional information |
|---|---|---|---|---|
| Antibody | anti-Dhx9 (Rabbit monoclonal) | ABclonal | Cat# A4563; RRID:AB_2863296 | IF(1:100), WB (1:1000) |
| Antibody | anti-GAPDH (Mouse monoclonal) | Proteintech | Cat# 60004–1-Ig; RRID:AB_2107436 | WB (1:200000) |
| Antibody | anti-Nanog (Rabbit monoclonal) | Proteintech | Cat#: 14295–1-AP; RRID:AB_1607719 | WB (1:2000) |
| Antibody | anti-Nanog (Mouse monoclonal) | Developmental Studies Hybridoma Bank (DHSB) | Cat#: PCRP-NANOGP1-2D8; RRID:AB_2722264 | IF (1:50) |
| Antibody | anti-Oct4 (Mouse monoclonal) | Santa Cruz | Cat#: sc-5279; RRID:AB_628051 | IF (1:200), WB (1:1000) |
| Antibody | anti-Lin28a (Rabbit monoclonal) | Proteintech | Cat#: 11724–1-AP; RRID:AB_2135039 | WB (1:1000) |
| Antibody | anti-rabbit IgG (H+L) HRP (Goat Polyclonal) | Bioss | Cat#: bs-0295G-HRP; RRID:AB_10923693 | WB (1:2000) |
| Antibody | anti-mouse IgG (H+L) HRP (Goat Polyclonal) | SinoBiological | Cat#: SSA007; RRID:AB_2917997 | WB (1:2000) |
| Antibody | anti-mouse IgG (H+L) Alexa Fluor 488 (Donkey Polyclonal) | Invitrogen | Cat#: A-21202; RRID:AB_141607 | IF (1:2000) |
| Recombinant protein | Streptavidin-Alexa647 | Bioss | Cat#: bs-0437P-AF647 | IF (1:200) |
| Sequence-based reagent | Dhx9 sgRNA1 | This paper | CRISPR sgRNA targeting Sequence | ATCAGAGGTGTCGCTAAGTA |
| Sequence-based reagent | Dhx9 sgRNA2 | This paper | CRISPR sgRNA targeting Sequence | GAAGGGGTTACCAGCACCAAT |

## Expression and purification of recombinant proteins

To obtain the plasmid expressing the recombinant protein GP41-pG-Tn5, we synthesized a DNA fragment encoding a single copy of GP41 peptide (*Boersma et al., 2019*) and then followed the Gibson assembly protocol (*Gibson et al., 2009*) to insert this fragment between 10xHis-tag and pG (GB1) of the plasmid pET28a_His-GB1-Tn5 (*Xu et al., 2021*) to get the plasmid pET28a_His-GP41-GB1-Tn5. For the plasmid expressing the recombinant protein HBD-V5, we synthesized the DNA fragment encoding 10xHis-tag, EGFP, two copies of HBD domain of RNase H1 with optimized codon, and 2xV5-tag (10xHis-EGFP-2xHBD-2xV5) and cloned it into pET28a to get the plasmid pET28a_His-EGFP-2xHBD-2xV5. For the plasmid expressing the recombinant protein BG4-V5, we synthesized the DNA fragment encoding BG4, 2xV5-tag, and 6xHis-tag (BG4-2xV5-His) and cloned it into pET28a to get the plasmid pET28a_ BG4-2xV5-His. For the plasmid expressing the recombinant protein mSA-anti-GP41scFv, we synthesized the DNA fragment encoding mSA (*Lim et al., 2013*), anti-GP41scFv (*Boersma et al., 2019*) and 10xHis-tag (mSA-GP41scFv-His) and cloned it into pET28a to get the plasmid pET28a_ mSA-GP41scFv-His.

All plasmids were transformed into *E. coli* BL21(DE3) cells. A single colony was cultured in a 5 mL LB medium containing 50 µg/ml kanamycin at 37 °C with 200 rpm shaking for overnight. The cells were then transferred to 500 mL of the LB medium for continued incubation at 37 °C until the optical density at 600 nm ($OD_{600}$) reached 0.8. IPTG was added to the final concentration of 0.2 mM for induction at 20 °C for 20 hr. Cells were collected and resuspended in Buffer A (50 mM Tris pH 7.4, 800 mM NaCl, and 20 mM imidazole). Cells were then lysed by sonication and centrifuged at 4 °C for 30 min at 12,000 rpm. Solubilized proteins in the supernatant were purified by the Ni-NTA resin (Cube biotech #31103) and eluted with Buffer B (50 mM Tris pH 7.4, 350 mM NaCl, and 500 mM imidazole). The eluted proteins were concentrated to remove imidazole and RNase through a 30 kDa cut-off Amicon Ultra centrifugal filter unit (Millipore #UFC903096). Glycerol was added to the concentrated protein to get the final concentration of 50%. The proteins with 50% glycerol were frozen at –80 °C. The purity of the recombinant proteins was analyzed by SDS-PAGE with Coomassie blue staining. Protein concentration was determined by OD280.

## Cell lines and cell culture

The HEK293 and mESC cell lines V6.5 were verified and obtained from Laurie Boyer's lab at the Massachusetts Institute of Technology (*Boyer et al., 2006*). Both the HEK293 and V6.5 cell lines are authenticated by the STR profiling and are negative for mycoplasma contamination testing by PCR method. HEK293 cells were maintained in the medium Dulbecco's Modified Eagle's Medium (DMEM, Sigma #D5671), 10% FBS (Gibco #10099–141), 1 x non-essential amino acid solution (Sigma #M7145), 1 x GlutaMAX (Sigma #G8541), 1 mM sodium pyruvate (Sigma #S8636), 0.1 mM β-mercaptoethanol (Sigma #M3148), 100 U/mL Penicillin-Streptomycin (Gibco #15140–122) under the condition of 37 °C and 5% $CO_2$. V6.5 cells were maintained on mitomycin C-inactivated mouse embryonic fibroblasts (MEFs) in the mESC medium DMEM, 10% FBS, 2000 U/ml LIF, 1 non-essential amino acid solution, 1x GlutaMAX, 1 mM sodium pyruvate, 0.1 mM β-mercaptoethanol, 100 U/mL Penicillin-Streptomycin under the condition of 37 °C and 5% $CO_2$. The cell lines were routinely checked free of mycoplasma contamination by PCR.

### Small molecule treatment

HEK293 cells were treated with DMSO or 25 µM ML216 (Selleck # S0469, inhibitor of BLM helicase) or 3 µM NSC 617145 (MCE # HY-110185, inhibitor of WRN helicase) for 16 hr, and then were collected for different assays.

## Generation of Dhx9[KO] ESCs with CRISPR/Cas9 genome editing

### Generation of Cas9 and sgRNAs expressing vectors

We synthesized the piggyBac 5' and 3' inverted repeat and cloned them into pUC57 to obtain pUC57.piggyBac. We then amplified the EF1a promoter (from Addgene# #26777), eSpCas9(1.1) (from Addgene #71814), and IRES-NeoR-WPRE cassette (from Addgene#50917) and ligated them in order to pUC57.piggyBac to get the PiggyBac_EF1a-eSpCas9-IRES-NeoR-WPRE by Gibson Assembly (*Gibson et al., 2009*). We synthesized the PGK promoter-PuroR-SV40 polyA cassette and two copies of the U6 promoter-guide RNA scaffold (one copy with two Bbs I sites and the other with two Bsa I

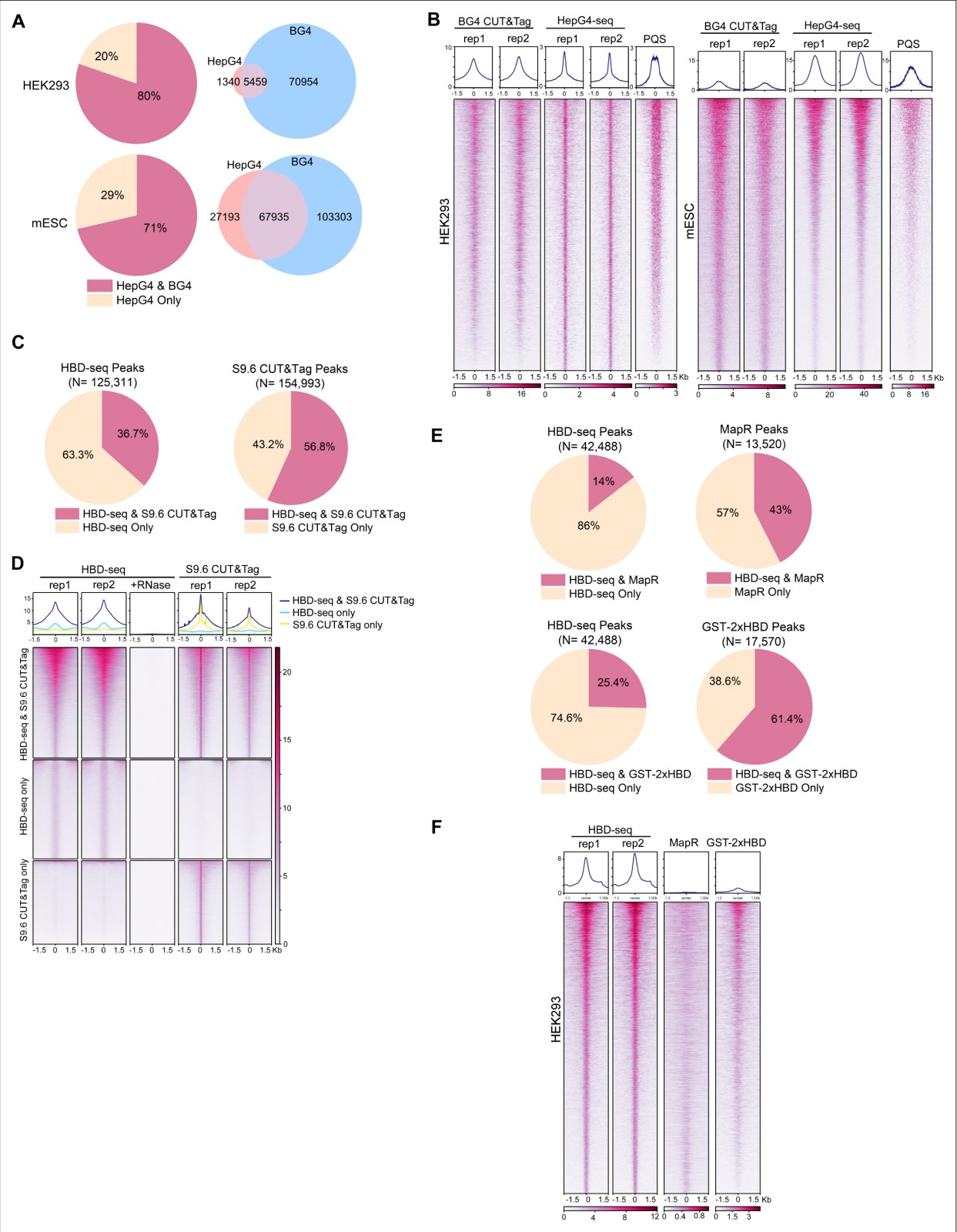

**Figure 9.** Comparisons of HepG4-seq and hybrid-binding domain (HBD)-seq with previous methods. (**A**) Pie charts showing the percentages of HepG4-seq peaks overlapping with peaks identified by BG4 CUT&Tag; Venn diagrams comparing HepG4-seq peaks and BG4 CUT&Tag seqs. (**B**) Heatmap showing the signal of BG4 CUT&Tag, HepG4-seq, and maxScores of PQS ±1.5 kb around the center of peaks identified by both BG4 CUT&Tag and HepG4-seq in HEK293 cells or mouse embryonic stem cells (mESCs). Color scales represent the density of the signals. Top: Profile plot showing the

*Figure 9 continued on next page*

Figure 9 continued

average signal. Rep1/rep2, two biologically independent replicates. (C) Pie charts showing the overlapping percentages of peaks between HBD-seq and S9.6 CUT&Tag in mESCs. (D) Heatmap showing the signal of HBD-seq and S9.6 CUT&Tag ± 1.5 kb around the center of peaks identified by HBD-seq and S9.6 CUT&Tag in mESCs. Color scales represent the density of the signals. Top: Profile plot showing the average signal. Rep1/rep2, two biologically independent replicates. (E) Pie charts showing the overlapping percentages of peaks between HBD-seq and MapR or GST-2xHBD CUT&Tag in HEK293 cells. (F) Heatmap showing the signal of HBD-seq, MapR, and GST-2xHBD CUT&Tag ±1.5 kb around the center of peaks identified by HBD-seq in HEK293 cells. Color scales represent the density of the signals. Top: Profile plot showing the average signal. Rep1/rep2, two biologically independent replicates.

sites for inserting CRISPR targeting sequence) and ligated them in order to pUC57 to obtain pUC57_sgRNAduo-Puro by Gibson Assembly (*Gibson et al., 2009*). Two sgRNAs were designed using http://crispor.tefor.net/. Two pairs of oligos for sgRNAs (Key resources table) were annealed and phosphorylated, and cloned into pUC57_sgRNAduo-Puro separately via Bbs I and Bsa I sites to obtain Dhx9_sgRNA1&2.

## Generation of dhx9$^{KO}$ ESCs

We transfected the PiggyBac_EF1a-eSpCas9-IRES-NeoR-WPRE plasmid and Supper PiggyBac Transposase plasmid (Beijing Zoman Biotech) into V6.5 mESCs using the Lonza Nucleofector 2b with the mouse ES cell nucleofector Kit and then treated the transfected cells with G418 (400 µg/ml) for five days and pick up clones individually under microscope to obtain the single clone constitutively expressing eSpCas9 (Cas9 mESCs), which was considered as wildtype cells in the case of comparing with *dhx9$^{KO}$* cells. Next, we transfected the plasmid Dhx9_sgRNA1&2 into Cas9 mESCs using the Lonza Nucleofector 2b with the mouse ES cell nucleofector Kit and treated cells with 1 µg/mL puromycin for two days 48 hr after transfection. The clones were picked up individually under a microscope and screened for *dhx9$^{KO}$* cells by western blot analysis using an anti-Dhx9 antibody (ABclonal, A4563).

## Western blot

mESCs were dissociated using trypsin and removed MEF cells, and then lysed in the RIPA buffer. Proteins in cell lysate were collected by centrifuging at 15,000 g for 15 min at 4 °C, and were separated by SDS-PAGE and electro-transferred to the 0.45 µm PVDF membrane (Millipore #IPVH00010). The membrane was blotted under 5% milk prepared by nonfat-dried milk and PBST (1 x PBS +0.1% Tween-20) and washed by PBST. The primary antibodies were incubated with membranes for overnight at 4 °C, and the HRP-conjugated secondary antibodies were incubated for 30 min at room temperature. Antibodies were listed in the key resources able.

## mESC differentiation into embryoid bodies

mESCs were cultured in mESC medium for two passages before differentiation. To initiate differentiation, cells were removed MEFs and diluted to 10,000 cells/mL in EB-Diff medium (standard mESCs medium without LIF). The 2.5 mL diluted cells were plated on the ultra-low-attachment six-well cell culture plate and resuspended cells by shaking the plate to avoid cell adherence. The medium was replaced by EB-Diff medium with 50 µg/mL ascorbic acid (Sigma #A4544) every other day. EBs were allowed to grow for 5 days and analyzed at various times for gene expression.

## RNA isolation and quantitative real-time PCR

Total RNAs were isolated by the acid guanidinium thiocyanate-phenol-chloroform extraction protocol from cells (*Chomczynski and Sacchi, 2006*). For RT-qPCR analysis, 1 µg total RNA was reverse-transcribed by MonScript RTIII All-in-One Mix (Monad, #MR05101M) following the instruction of the manufacturer. The RT product was subjected to real-time PCR using the SYBR Green PCR master mix and the Roche LC96 Lightcycler. Relative expression values were calculated by the ΔΔCT method using GAPDH for normalization. All reactions were performed in triplicate.

## Cell cycle analysis

Cells were fixed overnight at 4 °C in 100% ethanol, washed with PBS buffer containing 1% BSA, and stained with 1 µg/mL of DAPI and 1 mg/mL RNase A in PBS for 1 hr at 37 °C. Samples were analyzed on an Attune NxT flow cytometer. The cell cycle was analyzed using ModFit LT.

## Cell proliferation assay

Both WT and *dhx9*[KO] ESCs were seeded in 96-well cell culture plates at 5000, 7500, and 10,000 cells/well, respectively (six replicates per condition). Six blank wells were used as controls. Fresh medium was changed after 24 hr. After another 18 hr, cell proliferation was measured using the CCK-8 Cell Proliferation and Cytotoxicity Assay Kit (Solarbio, # CA1210).

## Immunofluorescence

Cells were fixed with 4% paraformaldehyde for 15 min at room temperature (*Figures 6 and 8*) or 100% cold methanol for 15 min at 4 °C (*Figure 1*). The fixed cells were washed three times with PBS and permeabilized in PBS containing 0.2% Triton-X100 and 0.1% Tween-20 for 10 min at room temperature. The permeabilized cells were washed twice with PBS and incubated in PBS containing 10% NBS and 0.1% Tween-20 for 1 hr at room temperature. Cells were stained with primary antibody overnight at 4 °C, and washed three times with PBS containing 0.1% Tween-20. Next, cells were stained with secondary antibody for 30 min at room temperature and washed three times with PBS containing 0.1% Tween-20. Finally, cells were stained with 10 ug/mL Hoechst33342 for 10 min at room temperature, and imaged using the Nikon Eclipse Ti2 microscope or Olympus SpinSR. Antibodies were listed in the key resources table.

## RNA sequencing (RNA-seq)

### RNA-seq

The mRNA was first isolated from 1 µg total RNAs using the poly(A) mRNA Capture Module (ABclonal, #RK20340). RNA-seq libraries were prepared from the isolated mRNAs using the Fast RNA-seq Lib Prep Kit V2 (ABclonal, #RK20306). The libraries were finally sequenced on NovaSeq 6000 platform (Illumina) by HaploX Genomics Center.

### Bioinformatic analysis

The pair-end (PE) sequencing reads were analyzed for quality control, filtered, and trimmed off adaptors to get the clean reads using fastp (*Chen et al., 2018*). Ribosomal RNAs (rRNAs) were removed from clean reads using SortMeRNA (*Kopylova et al., 2012*). Reads without rRNAs were mapped to the hg38 human genome or mm10 mouse genome using STAR (*Dobin et al., 2013*) with default settings to generate the bam files. RNA levels of each gene and biotypes of RNAs were quantified using RSEM (*Li and Dewey, 2011*) and normalized by the FPKM method. The bigwig file were converted from bam files by deepTools (*Ramírez et al., 2014*) (v2.0) and visualized using IGV (*Robinson et al., 2011*) genome browser. The differential gene expression analysis was performed using DESeq2 (*Love et al., 2014*).

## HepG4-seq, BG4-seq, HBD-seq, and Dhx9 CUT&Tag

### HepG4-seq

HEK293 cells or mESCs were treated with or without 25 µM Hemin (Non-label control) (Sigma, #51280) and 0.5 mM Biotin-aniline (Iris Biotech, #LS-3970) for 2 hr. Next, the $H_2O_2$ was added to a final concentration of 1 mM to activate the biotinylation reaction. After 1 min of very gentle shaking, all medium was replaced with quench buffer (1 x PBS, 10 mM Trolox,20 mM sodium ascorbate). After 2 min of very gentle shaking, cells were washed three times with cold PBS and dissociated by TrypLE digestion. Next, cells were washed twice with Buffer 1 (10 mM HEPES pH 7.5, 150 mM NaCl, 0.5 mM spermidine, 1x Protease inhibitor cocktail). Trypan blue was used to detect cell viability above 90%. Live cells ($1\times10^5$) were taken and conjugated to concanavalin A–coated magnetic beads (Epizyme Biotech, #YJ012) through incubation at room temperature for 15 min. The bead-bound cells were washed three times with Buffer 1 and resuspended in 50 µL AB buffer (Buffer 1 with 1% BSA, 2 mM EDTA, and 0.025% digitonin) containing 4.5 ug recombinant mSA-scFv at 4 °C by rotating overnight. Cells were then washed with DB-1 buffer (Buffer 1 with 0.025% digitonin) twice, and resuspended in 100 µL DB-300 buffer (Buffer 1 with 150 mM NaCl and 0.025% digitonin) containing 2 uL GP41-pG-Tn5 adapter complex and incubated at room temperature for 1 hr with rotation. To prepare the GP41-pG-Tn5 adapter complex, we incubated 12 µl 20 µM recombinant GP41-pG-Tn5 protein and 2.4 ul 100 µM annealed adaptor in 20 µl reaction as described (*Kaya-Okur et al., 2019*). Cells were

washed three times with DB-300 buffer to remove unbound GP41-pG-Tn5 protein. Next, biotinylated genomic DNA was fragmented by activating Tn5 with $Mg^{2+}$ and the sequencing library of biotinylated DNA fragments was built as described in the CUT&Tag study (*Kaya-Okur et al., 2019*). The library was finally sequenced on NovaSeq 6000 platform (Illumina) by HaploX Genomics Center.

## BG4-seq, HBD-seq

Cells were harvested using TrpLE and washed with Buffer 1. Live cells ($1\times10^5$) were taken and conjugated to concanavalin A–coated magnetic beads (Epizyme Biotech, #YJ012) through incubation at room temperature for 15 min. The bead-bound cells were washed three times with Buffer 1 and resuspended in 50 µL AB buffer containing 2 ug recombinant BG4-V5 (BG4-seq) or 3 µg recombinant HBD-V5 (HBD-seq) at 4 °C by rotating overnight. For the HBD-seq control samples, 1 uL of 20 mg/mL RNase A (NEB, #T3018-2) and 2 µL of 5 U/µL RNase H (NEB, #M0297S) were added to 50 µL of AB buffer and incubated at 37 °C for 4 hr with rotation prior to addition of the recombinant HBD-V5. After overnight incubation at 4 °C, cells were washed once with DB-1 buffer and resuspended in 50 µL DB-1 buffer containing 2 ug anti-V5 (Sino Biological, #100378-T36). After incubation at room temperature for 1 hr with rotation, cells were washed twice with DB-1 buffer and resuspended in 50 µL DB-1 buffer containing 0.25 ug anti-Rabbit IgG (Sino Biological, #SSA018). After incubation at room temperature for 30 min with rotation, cells were washed twice with DB-1 buffer and tagmented with GP41-pG-Tn5 adapter complex as HepG4-seq. The sequencing library was prepared as CUT&Tag (*Kaya-Okur et al., 2019*) using the tagmented fragments. The library was finally sequenced on NovaSeq 6000 platform (Illumina) by HaploX Genomics Center.

## Dhx9 CUT&Tag

Cells were harvested using TrpLE and washed with Buffer 1. Live cells ($1\times10^5$) were taken and conjugated to concanavalin A–coated magnetic beads (Epizyme Biotech, #YJ012) through incubation at room temperature for 15 min. Cells were fixed with 0.1% formaldehyde (Cell Signaling, #50-00-0) for 2 min as described (*Kaya-Okur et al., 2019*). The bead-bound cells were washed three times with Buffer 1 and resuspended in 50 µL AB buffer containing 0.04 ug Dhx9 antibody (ABclonal, #A4563) at 4 °C by rotating overnight. After overnight incubation at 4 °C, cells were washed once with DB-1 buffer and resuspended in 50 µL DB-1 buffer 0.25 ug anti-Rabbit IgG (Sino Biological, #SSA018). After incubation at room temperature for 30 mins with rotation, cells were washed twice with DB-1 buffer and tagmented with pG-Tn5 adapter complex (*Xu et al., 2021*). The sequencing library was prepared as CUT&Tag (*Kaya-Okur et al., 2019*) using the tagmented fragments. The library was finally sequenced on NovaSeq 6000 platform (Illumina) by HaploX Genomics Center.

## Bioinformatic analysis

The PE sequencing reads were analyzed for quality control, filtered, and trimmed off adaptors to get the clean reads using fastp (*Chen et al., 2018*) and Cutadapt (*Martin, 2011*). Clean Reads were mapped to hg38 human genome or mm10 mouse genome using Bowie2 (*Langmead and Salzberg, 2012*) with the settings (`--end-to-end --very-sensitive --no-mixed --no-discordant --phred33 -I 10 X 500`) to generate the bam files. High-quality mapping reads were extracted by setting parameter samtools -q to 40 (*Cheng et al., 2009*). Duplicate reads were removed using the Picard toolkit (RRID:SCR_006525, https://broadinstitute.github.io/picard/). Only uniquely mapping and de-duplicated reads (quality score >40) were retained. To normalize reads coverage, scale factors were calculated by ChIPseqSpikeInFree software (*Jin et al., 2020*) with default parameters. Given the difference between DMSO and inhibitor treatment is too big, ChIPseqSpikeInFree failed to calculate the scale factor and thereby we calculated the scale factor using the total numbers of uniquely mapping reads. To create coverage plots or heatmaps, the reads coverage were normalized by setting the scaleFactor parameters and converted to the bigwig files from bam files using deepTools (*Ramírez et al., 2014*). The heatmaps and profile plots were generated from normalized bed files using deepTools (*Ramírez et al., 2014*). The bigwig files were visualized using the IGV genome browser (*Robinson et al., 2011*). The bam files were converted to normalized bedgraph files using the BEDTools genomecov command and the scale factor calculated above. Peaks were called from normalized bedgraph files using SEACR (*Meers et al., 2019*) with the settings (0.05 non-stringent). High confident peaks were consensus peaks between two replicates generated with BEDTools (*Quinlan*

*and Hall, 2010*) intersect command. Co-localized G4s and R-loops peaks are overlapping G4 and DNA/RNA hybrid peaks generated with BEDTools (*Quinlan and Hall, 2010*) intersect command. To detect the differential peaks, peaks from different groups were first re-centered and trimmed based on calculating their reads coverage within a fix bin (400 bp) using R package DiffBind (v3.12); the read coverage of each peak was normalized using the scale factor calculated above; the significantly differential peaks were defined as peaks with the significance cut-off p-value <0.05 and fold change (FC) ≥1.5; BEDTools were then used to identify overlaps between the significantly differential peaks and original peaks from SEACR to retrieve original peak sites with differential levels in different groups that were used subsequently as input file of HOMER software (*Heinz et al., 2010*). Genomic localization of peaks was annotated by HOMER annotatePeaks.pl (*Heinz et al., 2010*). Motif enrichment of peaks was analyzed by HOMER findMotifsGenome.pl (*Heinz et al., 2010*). G4/DNA/RNA hybrid/ Co-localized G4s and R-loops-associated genes are the nearest genes annotated by HOMER to G4/ DNA/RNA hybrid/Co-localized G4s and R-loops peaks. MapR, GST-2xHBD CUT&Tag, S9.6 CUT&Tag were analyzed using the same pipeline above.

## Analysis of histone marks, promoters, and enhancers

The bioinformatic analysis pipeline of these ChIP-seq data is the same as the HepG4-seq except that ChIP-seq peaks were called from bam files using MACS2 with default settings and the input DNAs were used as the control for MACS2. For histone marks, promoters are defined as a 2 kb region centered on the TSS of each Ensemble transcript as *Wamstad et al., 2012*. The enhancers was generated by combining all H3K4me1 or H3K27ac enriched regions and further filtered by removing any interval that overlapped with regions +/-1 kb of TSS annotated by mm10 genome build as *Wamstad et al., 2012*. If an extended region overlapped the +/-1 kb window around a TSS, the region was trimmed +/-5 kb of the TSS, leaving the distal portion in either direction as independent enhancer regions (*Wamstad et al., 2012*).

## GO enrichment analysis and STRING protein-protein interaction analysis

GO enrichment analysis was performed using the Metascape web interface (*Zhou et al., 2019b*) and G4/DNA/RNA hybrid/Co-localized G4s and R-loops-associated genes' ID as input. The STRING protein-protein physical interaction analysis was performed using the STRING web interface (*Szklarczyk et al., 2023*) with the settings (physical subnetwork, network edges indicate confidence, medium confidence 0.4).

## Acknowledgements

We thank Laurie Boyer at the Massachusetts Institute of Technology for providing the HEK293 and V6.5 cell lines. We thank Yi Liu (University of Texas Southwestern Medical Center, USA) and Fan Lai (Yunnan University, China) for insightful discussions and for critical evaluation of the manuscript. Z X is supported by the National Key Research and Development Program of China, Stem Cell and Translational Research (2018YFA0109200), and Project for Cultivating Leading Talent in Science and Technology of Sichuan University (0082604151357). LY is supported by the Postdoctoral Research Foundation of China (2018M643467).

## Additional information

### Funding

| Funder | Grant reference number | Author |
| --- | --- | --- |
| National Key Research and Development Program of China Stem Cell and Translational Research | 2018YFA0109200 | Zhihong Xue |

| Funder | Grant reference number | Author |
| --- | --- | --- |
| Project for Cultivating Leading Talent in Science and Technology of Sichuan University | 0082604151357 | Zhihong Xue |
| Postdoctoral Research Foundation of China | 2018M643467 | Lin Yu |

The funders had no role in study design, data collection and interpretation, or the decision to submit the work for publication.

### Author contributions

Ting Liu, Formal analysis, Validation, Investigation, Visualization, Methodology, Writing – original draft; Xing Shen, Data curation, Formal analysis, Validation, Investigation, Visualization, Methodology, Writing – original draft; Yijia Ren, Data curation, Formal analysis, Visualization, Writing – review and editing; Hongyu Lu, Data curation, Investigation; Yu Liu, Chong Chen, Writing – review and editing; Lin Yu, Conceptualization, Data curation, Supervision, Funding acquisition, Investigation, Visualization, Methodology, Writing – original draft, Project administration, Writing – review and editing; Zhihong Xue, Conceptualization, Resources, Data curation, Formal analysis, Supervision, Funding acquisition, Investigation, Visualization, Methodology, Writing – original draft, Project administration, Writing – review and editing

### Author ORCIDs

Chong Chen https://orcid.org/0000-0002-6787-0495
Zhihong Xue https://orcid.org/0000-0003-3568-2964

Reviewer #1 (Public review): https://doi.org/10.7554/eLife.99026.3.sa1
Reviewer #2 (Public review): https://doi.org/10.7554/eLife.99026.3.sa2
Reviewer #3 (Public review): https://doi.org/10.7554/eLife.99026.3.sa3
Author response https://doi.org/10.7554/eLife.99026.3.sa4

---

# Additional files

### Supplementary files

• Supplementary file 1. G-quadruplex (G4) and R-loop peaks in HEK293 cells. (a) HepG4-seq peaks in HEK293 cells (b) Merged HepG4-seq peaks in HEK293 cells treated with DMSO, ML216, NSC617145 (c) HBD-seq peaks in HEK293 cells (d) Co-localized G4 and R-loop peaks in HEK293 cells € Co-localized G4 and R-loop peaks-associated genes with differential expression after treatment with G4 Inhibitor.

• Supplementary file 2. G-quadruplex (G4) and R-loop peaks in mouse embryonic stem cells (mESCs). (a) HepG4-seq peaks in mESCs (b) HBD-seq peaks in mESCs (c) Co-localized G4s and R-loops in mESCs (d) Co-localized G4s and R-loops in the promoters of mESCs (e) Co-localized G4s and R-loops in the enhancers of mESCs (f) Differential G4 and R-loop peaks in dhx9KO mESCs compared to WT mESCs (g) Genes with differential expression levels in dhx9KO mESCs compared to wild-type (WT) mESCs (h) Dhx9 CUT&Tag peaks in mESCs (i) Co-localized G4s and R-loops bound by Dhx9 in mESCs.

• MDAR checklist

### Data availability

The HepG4-seq, HBD-seq, BG4-seq, Dhx9 CUT&Tag and RNA-seq data have been deposited to the Gene Expression Omnibus (accession code GSE254764 and GSE254763). The ChIP-seq data of histone markers and RNAP are openly available in GEO (GSE47950; *Wamstad et al., 2012*). Data of BG4 CUT&Tag in mESCs, S9.6 CUT&Tag in mESCs, GST-2xHBD CUT&Tag in HEK293 cells and MapR in HEK293 cells are openly available in GEO (GSE173103 and GSE156400).

The following datasets were generated:

| Author(s) | Year | Dataset title | Dataset URL | Database and Identifier |
|---|---|---|---|---|
| Xue Z | 2024 | Genome-wide mapping of native G-loops in living cells | https://www.ncbi.nlm.nih.gov/geo/query/acc.cgi?acc=GSE254764 | NCBI Gene Expression Omnibus, GSE254764 |
| Xue Z | 2024 | RNA-seq analysis of wild-type HEK293, HEK293 treated with DMSO, ML216 or NSC617145, wild-type mESCs and dhx9KO mESCs | https://www.ncbi.nlm.nih.gov/geo/query/acc.cgi?acc=GSE254763 | NCBI Gene Expression Omnibus, GSE254763 |

The following previously published datasets were used:

| Author(s) | Year | Dataset title | Dataset URL | Database and Identifier |
|---|---|---|---|---|
| Alexander J, Wamstad J | 2013 | Dynamic and Coordinated Epigenetic Regulation of Developmental Transitions in the Cardiac Lineage | https://www.ncbi.nlm.nih.gov/geo/query/acc.cgi?acc=GSE47950 | NCBI Gene Expression Omnibus, GSE47950 |
| Lyu J, Elsässer SJ | 2021 | Genome-wide mapping of G-quadruplex structures with CUT&Tag | https://www.ncbi.nlm.nih.gov/geo/query/acc.cgi?acc=GSE173103 | NCBI Gene Expression Omnibus, GSE173103 |
| Liang K, Fang P | 2021 | Genomic Profiling of Native R-Loops with a DNA-RNA Hybrid Recognition Sensor Protein | https://www.ncbi.nlm.nih.gov/geo/query/acc.cgi?acc=GSE156400 | NCBI Gene Expression Omnibus, GSE156400 |

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
