## [Editor Report · eLife assessment]

This study describes a **useful** antibody-free method to map both G-quadruplexes and R-loops in vertebrate cells independently of the BG4 and S9.6 antibodies. It also reveals that the helicase Dhx9 can affect the self-renewal and differentiation capacities of mESCs, perhaps by regulating co-localized G4s and R-loops. The datasets provided might constitute a good starting point for future functional studies, and although the strength of the evidence that DHX9 interferes with the ability of mESCs to differentiate by regulating directly the stability of either G4s or R-loops has been improved compared to a previous version, it is still **incomplete**.

---

## [Referee Report · Reviewer #1 (Public review)]

This study describes a useful antibody-free method to map G-quadruplexes in vertebrate cells. The analysis of the data is solid but it remains primarily descriptive and does not substantially add to existing publications (such as PMID:34792172 for example). Nevertheless, the datasets generated here might constitute a good starting point for more functional studies.

Comments on revised version:

It is disappointing to see that the authors decided to brush aside most of the comments made by the three referees, even though these comments were largely consistent with each other. As a result, the revised manuscript is not substantially changed or improved. Legitimate concerns regarding the specificity of the Cut&Tag signals were not addressed and therefore remain. The sensitivity of the HBD-seq signals to a combination of RNase A and RNase H does not demonstrate that HBD-seq specifically reports the presence of RNA:DNA hybrids. The new Figure 9 comparing HepG4-seq to existing datasets does not unequivocally demonstrate the superiority of the Hemin-based strategy to map G4s.

---

## [Referee Report · Reviewer #2 (Public review)]

Summary:

In this study, Liu et al. explore the interplay between G-quadruplexes (G4s) and R-loops. The authors developed novel techniques, HepG4-seq and HBD-seq, to capture and map these nucleic acid structures genome-wide in human HEK293 cells and mouse embryonic stem cells (mESCs). They identified dynamic, cell-type-specific distributions of co-localized G4s and R-loops, which predominantly localize at active promoters and enhancers of transcriptionally active genes. Furthermore, they assessed the role of helicase Dhx9 in regulating these structures and their impact on gene expression and cellular functions.

The manuscript provides a detailed catalogue of the genome-wide distribution of G4s and R-loops. However, the conceptual advance and the physiological relevance of the findings are not obvious. Overall, the impact of the work on the field is limited to the utility of the presented methods and datasets.

Strengths:

(1) The development and optimization of HepG4-seq and HBD-seq offer novel methods to map native G4s and R-loops.

(2) The study provides extensive data on the distribution of G4s and R-loops, highlighting their co-localization in human and mouse cells.

(3) The study consolidates the role of Dhx9 in modulating these structures and explores its impact on mESC self-renewal and differentiation.

Comments on revised version:

In this revised manuscript, Liu et al. address most of the previous concerns raised by this reviewer. Namely, the comparison between the novel methods and existing ones is an important addition.

---

## [Referee Report · Reviewer #3 (Public review)]

Summary:

The authors developed and optimized the methods for detecting G4s and R-loops independent of BG4 and S9.6 antibody, and mapped genomic native G4s and R-loops by HepG4-seq and HBD-seq, revealing that co-localized G4s and R-loops participate in regulating transcription and affecting the self-renewal and differentiation capabilities of mESCs.

Strengths:

By utilizing the peroxidase activity of G4-hemin complex and combining proximity labeling technology, the authors developed HepG4-seq (high throughput sequencing of hemin-induced proximal labelled G4s) , which can detect the dynamics of G4s in vivo. Meanwhile, the "GST-His6-2xHBD"-mediated CUT&Tag protocol (Wang et al., 2021) was optimized by replacing fusion protein and tag, the optimized HBD-seq avoids the generation of GST fusion protein aggregates and can reflect the genome-wide distribution of R-loops in vivo.

The authors employed HepG4-seq and HBD-seq to establish comprehensive maps of native co-localized G4s and R-loops in human HEK293 cells and mouse embryonic stem cells (mESCs). The data indicate that co-localized G4s and R-loops are dynamically altered in a cell type-dependent manner and are largely localized at active promoters and enhancers of transcriptional active genes.

Combined with Dhx9 ChIP-seq and co-localized G4s and R-loops data in wild-type and dhx9KO mESCs, the authors found that the helicase Dhx9, a major regulator of co-localized G4s and R-loops, affects the self-renewal and differentiation capacities of mESCs.

In conclusion, the authors provide an approach to study the interplay between G4s and R-loops, shedding light on the important roles of co-localized G4s and R-loops in development and disease by regulating the transcription of related genes.

Weaknesses:

As we know, there are at least two structure data of S9.6 antibody very recently, and the questions about the specificity of the S9.6 antibody on RNA:DNA hybrids should be finished. The authors referred (Hartono et al., 2018; Konig et al., 2017; Phillips et al., 2013) need to be updated, and the author's bias against S9.6 antibodies needs also to be changed. In contrast to S9.6 CUT&Tag and other inactive ribonucleotide H1-based methods including MapR (inactive ribonucleotide H1-mediated CUT&Run) (Yan et al., 2019)and GST-2xHBD CUT&Tag (Wang et al., 2021), HBD-seq did not perform satisfactorily and its binding specificity was questionable.

Although HepG4-seq is an effective G4s detection technique, and the authors have also verified its reliability to some extent, given the strong link between ROS homeostasis and G4s formation, hemin's affinity for different types of G4s and their differences in peroxidase activities, whether HepG4-seq reflects the dynamics of G4s in vivo more accurately than existing detection techniques still needs to be more carefully corroborated.

The authors focus on the interaction of non-B DNA structures G4s and R-loops and their roles in development and disease by regulating the transcription of related genes. Compared to the complex regulatory network of G4s and R-loops, the authors provide limited mechanistic insight into the major regulator of co-localized G4s and R-loops, helicase Dhx9. However, the authors propose that "A degron system-mediated simultaneous and/or stepwise degradation system of multiple regulators will help us elucidate the interplaying effects between G4s and R-loops." is attractive. The main innovations of this article are the proposal of new antibody-independent methods for detecting G4s and the optimization of the GST-2xHBD CUT&Tag (Wang et al., 2021) method for detecting R-loops. Unfortunately, however, the reliability and accuracy of these methods are still debatable, and the reference value of the G4s and R-loops datasets based on these methods is relatively limited.

---

## [Author Response]

The following is the authors’ response to the original reviews.

**eLife assessment**
This useful study describes an antibody-free method to map G-quadruplexes (G4s) in vertebrate cells. While the method might have potential, the current analysis is primarily descriptive and does not add substantial new insights beyond existing data (e.g., PMID:34792172). While the datasets provided might constitute a good starting point for future functional studies, additional data and analyses would be needed to fully support the major conclusions and, at the same time, clarify the advantage of this method over other methods. Specifically, the strength of the evidence for DHX9 interfering with the ability of mESCs to differentiate by regulating directly the stability of either G4s or R-loops is still incomplete.

We thank the editors for their helpful comments.

Given that antibody-based methods have been reported to leave open the possibility of recognizing partially folded G4s and promoting their folding, we have employed the peroxidase activity of the G4-hemin complex to develop a new method for capturing endogenous G4s that significantly reduces the risk of capturing partially folded G4s. We have included a new Fig. 9 and a new section “Comparisons of HepG4-seq and HBD-seq with previous methods” to carefully compare our methods to other methods.

In the Fig. 7, we applied the Dhx9 CUT&Tag assay to identify the G4s and R-loops directly bound by Dhx9 and further characterized the differential Dhx9-bound G4s and R-loops in the absence of Dhx9. Dhx9 is a versatile helicase capable of directly resolving R-loops and G4s or promoting R-loop formation (PMID: 21561811, 30341290, 29742442, 32541651, 35905379, 34316718). Furthermore, we showed that depletion of Dhx9 significantly altered the levels of G4s or R-loops around the TSS or gene bodies of several key regulators of mESC and embryonic development, such as Nanog, Lin28a, Bmp4, Wnt8a, Gata2, and Lef1, and also their RNA levels (Fig.7 I). The above evidence is sufficient to support the transcriptional regulation of mESCs cell fate by directly modulating the G4s or R-loops within the key regulators of mESCs.

**Public Reviews:**

**Reviewer #1 (Public Review):**
Summary:Non-B DNA structures such as G4s and R-loops have the potential to impact genome stability, gene transcription, and cell differentiation. This study investigates the distribution of G4s and R-loops in human and mouse cells using some interesting technical modifications of existing Tn5-based approaches. This work confirms that the helicase DHX9 could regulate the formation and/or stability of both structures in mouse embryonic stem cells (mESCs). It also provides evidence that the lack of DHX9 in mESCs interferes with their ability to differentiate.Strengths:HepG4-seq, the new antibody-free strategy to map G4s based on the ability of Hemin to act as a peroxidase when complexed to G4s, is interesting. This study also provides more evidence that the distribution pattern of G4s and R-loops might vary substantially from one cell type to another.

We appreciate your valuable points.

Weaknesses:This study is essentially descriptive and does not provide conclusive evidence that lack of DHX9 does interfere with the ability of mESCs to differentiate by regulating directly the stability of either G4 or R-loops. In the end, it does not substantially improve our understanding of DHX9's mode of action.

In this study, we aimed to report new methods for capturing endogenous G4s and R-loops in living cells. Dhx9 has been reported to directly unwind R-loops and G4s or promote R-loop formation (PMID: 21561811, 30341290, 29742442, 32541651, 35905379, 34316718). To understand the direct Dhx9-bound G4s and R-loops, we performed the Dhx9 CUT&Tag assay and analyzed the co-localization of Dhx9-binding sites and G4s or R-loops. We found that 47,857 co-localized G4s and R-loops are directly bound by Dhx9 in the wild-type mESCs and 4,060 of them display significantly differential signals in absence of Dhx9, suggesting that redundant regulators exist as well. We showed that depletion of Dhx9 significantly altered the RNA levels of several key regulators of mESC and embryonic development, such as Nanog, Lin28a, Bmp4, Wnt8a, Gata2, and Lef1, which coincides with the significantly differential levels of G4s or R-loops around the TSS or gene bodies of these genes (Fig.7). The comprehensive molecular mechanism of Dhx9 action is indeed not the focus of this study. We will work on it in the future studies. Thank you for the comments.

There is no in-depth comparison of the newly generated data with existing datasets and no rigorous control was presented to test the specificity of the hemin-G4 interaction (a lot of the hemin-dependent signal seems to occur in the cytoplasm, which is unexpected).

The specificity of hemin-G4-induced peroxidase activity and self-biotinylation has been well demonstrated in previous studies (PMID: 19618960, 22106035, 28973477, 32329781). In the Fig.1A, we compared the hemin-G4-induced biotinylation levels in different conditions. Cells treated with hemin and Bio-An exhibited a robust fluorescence signal, while the absence of either hemin or Bio-An almost completely abolished the biotinylation signals, suggesting a specific and active biotinylation activity. To identify the specific signals, we have included the non-label control and used this control to call confident HepG4 peaks in all HepG4-seq assays.

The hemin-RNA G4 complex has also been reported to have mimic peroxidase activity and trigger similar self-biotinylation signals as DNA G4s (PMID: 32329781, 31257395, 27422869). Therefore, it is not surprising to observe hemin-dependent signals in the cytoplasm generated by cytoplasmic RNA G4s.

In the revised version, we have included a new Fig. 9 and a new section “Comparisons of HepG4-seq and HBD-seq with previous methods” to carefully compare our methods to other methods.

The authors talk about co-occurrence between G4 and R-loops but their data does not actually demonstrate co-occurrence in time. If the same loci could form alternatively either R-loops or G4 and if DHX9 was somehow involved in determining the balance between G4s and R-loops, the authors would probably obtain the same distribution pattern. To manipulate R-loop levels in vivo and test how this affects HEPG4-seq signals would have been helpful.

Single-molecule fluorescence studies have shown the existence of a positive feedback mechanism of G4 and R-loop formation during transcription (PMID: 32810236, 32636376), suggesting that G4s and Rloops could co-localize at the same molecule. Dhx9 is a versatile helicase capable of directly resolving R-loops and G4s or promoting R-loop formation (PMID: 21561811, 30341290, 29742442, 32541651, 35905379, 34316718). Although depletion of Dhx9 resulted in 6,171 Dhx9-bound co-localized G4s and R-loops with significantly altered levels of G4s or R-loops, only 276 of them (~4.5%) harbored altered G4s and R-loops, suggesting that the interacting G4s and R-loops are rare in living cells. Nowadays, the genome-wide co-occurrence of two factors are mainly obtained by bioinformatically intersection analysis. We agreed that F We will carefully discuss this point in the revised version. At the same time, we will make efforts to develop a new method to map the co-localized G4 and R-loop in the same molecule in the future study.

This study relies exclusively on Tn5-based mapping strategies. This is a problem as global changes in DNA accessibility might strongly skew the results. It is unclear at this stage whether the lack of DHX9, BLM, or WRN has an impact on DNA accessibility, which might underlie the differences that were observed. Moreover, Tn5 cleaves DNA at a nearby accessible site, which might be at an unknown distance away from the site of interest. The spatial accuracy of Tn5-based methods is therefore debatable, which is a problem when trying to demonstrate spatial co-occurrence. Alternative mapping methods would have been helpful.

In this study, we used the recombinant streptavidin monomer and anti-GP41 nanobody fusion protein (mSA-scFv) to specifically recognize hemin-G4-induced biotinylated G4 and then recruit the recombinant GP41-tagged Tn5 protein to these G4s sites. Similarly, the recombinant V5-tagged N-terminal hybrid-binding domain (HBD) of RNase H1 specifically recognizes R-loops and recruit the recombinant protein G-Tn5 (pG-Tn5) with the help of anti-V5 antibody. Therefore, the spatial distance of Tn5 to the target sites is well controlled and very short, and also the recruitment of Tn5 is specifically determined by the existence of G4s in HepG4-seq and R-loops in HBD-seq. In addition, RNase treatment markedly abolished the HBD-seq signals and the non-labeled controls exhibit obviously reduction of HepG4-seq signals, demonstrating that HBD-seq and HepG4-seq were not contamination from tagmentation of asccessible DNA.

**Reviewer #2 (Public Review):**
Summary:In this study, Liu et al. explore the interplay between G-quadruplexes (G4s) and R-loops. The authors developed novel techniques, HepG4-seq and HBD-seq, to capture and map these nucleic acid structures genome-wide in human HEK293 cells and mouse embryonic stem cells (mESCs). They identified dynamic, cell-type-specific distributions of co-localized G4s and R-loops, which predominantly localize at active promoters and enhancers of transcriptionally active genes. Furthermore, they assessed the role of helicase Dhx9 in regulating these structures and their impact on gene expression and cellular functions.The manuscript provides a detailed catalogue of the genome-wide distribution of G4s and R-loops. However, the conceptual advance and the physiological relevance of the findings are not obvious. Overall, the impact of the work on the field is limited to the utility of the presented methods and datasets.Strengths:(1) The development and optimization of HepG4-seq and HBD-seq offer novel methods to map native G4s and R-loops.(2) The study provides extensive data on the distribution of G4s and R-loops, highlighting their co-localization in human and mouse cells.(3) The study consolidates the role of Dhx9 in modulating these structures and explores its impact on mESC self-renewal and differentiation.

We appreciate your valuable points.

Weaknesses:(1) The specificity of the biotinylation process and potential off-target effects are not addressed. The authors should provide more data to validate the specificity of the G4-hemin.

The specificity of hemin-G4-induced peroxidase activity and self-biotinylation has been well demonstrated in previous studies (PMID: 19618960, 22106035, 28973477, 32329781). In the Fig.1A, we compared the hemin-G4-induced biotinylation levels in different conditions. Cells treated with hemin and Bio-An exhibited a robust fluorescence signal, while the absence of either hemin or Bio-An almost completely abolished the biotinylation signals, suggesting a specific and active biotinylation activity.

(2) Other methods exploring a catalytic dead RNAseH or the HBD to pull down R-loops have been described before. The superior quality of the presented methods in comparison to existing ones is not established. A clear comparison with other methods (BG4 CUT&Tag-seq, DRIP-seq, R-CHIP, etc) should be provided.

Thank you for the suggestions. We have included a new Fig. 9 and a new section “Comparisons of HepG4-seq and HBD-seq with previous methods” to carefully compare our methods to other methods.

(3) Although the study demonstrates Dhx9's role in regulating co-localized G4s and R-loops, additional functional experiments (e.g., rescue experiments) are needed to confirm these findings.

Dhx9 has been demonstrate as a versatile helicase capable of directly resolving R-loops and G4s or promoting R-loop formation in previous studies (PMID: 21561811, 30341290, 29742442, 32541651, 35905379, 34316718). We believe that the current new dataset and previous studies are enough to support the capability of Dhx9 in regulating co-localized G4s and R-loops.

(4) The manuscript would benefit from a more detailed discussion of the broader implications of co-localized G4s and R-loops.

Thank you for the suggestions. We have included the discussion in the revised version.

(5) The manuscript lacks appropriate statistical analyses to support the major conclusions.

We apologized for this point. Whereas we have applied careful statistical analyses in this study, lacking of some statistical details make people hard to understand some conclusions. We have carefully added details of all statistical analysis.

(6) The discussion could be expanded to address potential limitations and alternative explanations for the results.

Thank you for the suggestions. We have included the discussion about this point in the revised version.

**Reviewer #3 (Public Review):**
Summary:The authors developed and optimized the methods for detecting G4s and R-loops independent of BG4 and S9.6 antibody, and mapped genomic native G4s and R-loops by HepG4-seq and HBD-seq, revealing that co-localized G4s and R-loops participate in regulating transcription and affecting the self-renewal and differentiation capabilities of mESCs.Strengths:By utilizing the peroxidase activity of G4-hemin complex and combining proximity labeling technology, the authors developed HepG4-seq (high throughput sequencing of hemin-induced proximal labelled G4s), which can detect the dynamics of G4s in vivo. Meanwhile, the "GST-His6-2xHBD"-mediated CUT&Tag protocol (Wang et al., 2021) was optimized by replacing fusion protein and tag, the optimized HBD-seq avoids the generation of GST fusion protein aggregates and can reflect the genome-wide distribution of R-loops in vivo.The authors employed HepG4-seq and HBD-seq to establish comprehensive maps of native co-localized G4s and R-loops in human HEK293 cells and mouse embryonic stem cells (mESCs). The data indicate that co-localized G4s and R-loops are dynamically altered in a cell type-dependent manner and are largely localized at active promoters and enhancers of transcriptionally active genes.Combined with Dhx9 ChIP-seq and co-localized G4s and R-loops data in wild-type and dhx9KO mESCs, the authors confirm that the helicase Dhx9 is a direct and major regulator that regulates the formation and resolution of co-localized G4s and R-loops.Depletion of Dhx9 impaired the self-renewal and differentiation capacities of mESCs by altering the transcription of co-localized G4s and R-loops-associated genes.In conclusion, the authors provide an approach to studying the interplay between G4s and R-loops, shedding light on the important roles of co-localized G4s and R-loops in development and disease by regulating the transcription of related genes.

We appreciate your valuable points.

Weaknesses:As we know, there are at least two structure data of S9.6 antibody very recently, and the questions about the specificity of the S9.6 antibody on RNA:DNA hybrids should be finished. The authors referred to (Hartono et al., 2018; Konig et al., 2017; Phillips et al., 2013) need to be updated, and the authors' bias against S9.6 antibodies needs also to be changed. However, as the authors had questioned the specificity of the S9.6 antibody, they should compare it in parallel with the data they have and the data generated by the widely used S9.6 antibody.

Thank you for the updating information about the structure data of S9.6 antibody. We politely disagree the specificity of the S9.6 antibody on RNA:DNA hybrids. The structural studies of S9.6 (PMID: 35347133, 35550870) used only one RNA:DNA hybrid to show the superior specificity of S9.6 on RNA:DNA hybrid than dsRNA and dsDNA. However, Fabian K. et al has reported that the binding affinities of S9.6 on RNA:DNA hybrid exhibits obvious sequence-dependent bias from null to nanomolar range (PMID: 28594954). We have included the comparison between S9.6-derived data and our HBD-seq data in the Fig.9 and the section “Comparisons of HepG4-seq and HBD-seq with previous methods”.

Although HepG4-seq is an effective G4s detection technique, and the authors have also verified its reliability to some extent, given the strong link between ROS homeostasis and G4s formation, and hemin's affinity for different types of G4s, whether HepG4-seq reflects the dynamics of G4s in vivo more accurately than existing detection techniques still needs to be more carefully corroborated.

Thank you for pointing out this issue. In the in vitro hemin-G4 induced self-biotinylation assay, parallel G4s exhibit higher peroxidase activities than anti-parallel G4s. Thus, the dynamics of G4 conformation could affect the HepG4-seq signals (PMID: 32329781). In the future, people may need to combine HepG4-seq and BG4s-eq to carefully explain the endogenous G4s. We have discussed this point in the revised version.

**Recommendations for the authors:**

**Reviewer #1 (Recommendations For The Authors):**
Figures 1A&1G. Although no merge images were provided, it seems that the biotin signals are strongly enriched outside the nucleus. This suggests that hemin is not specific for G4s in DNA. Does it mean that Hemin can also recognise G4 on RNAs? How do the authors understand the cytoplasmic signal?

Hemin indeed could interact with RNA G4 to obtain the peroxidase activity like DNA G4-hemin complex (PMID: 27422869, 32329781, 31257395). The cytoplasmic signals in Figure 1A&1G were derived from RNA G4.

Figure 1A: The fact that there is no Alexa647 signal without hemin or Bio-An does not actually demonstrate that the signals are specific. These controls do not actually test for the specificity of the G4-Hemin interaction.

The specificity of hemin-G4-induced peroxidase activity and self-biotinylation has been well demonstrated in previous studies (PMID: 19618960, 22106035, 28973477, 32329781). In this study, we performed the IF to confirm this phenomena.

Figure 1C: It looks like the HepG4-seq signals are simply an amplification of the noise given by the Tn5 (the non-label ctrl has the same pattern, albeit weaker). It is unclear why this happens but it might happen if somehow hemin increased the probability that the Tn5 is close to chromatin in an unspecific manner (it would cut G-rich, nucleosome-poor, accessible sites in an unspecific manner). To discard this possibility, it would be interesting to investigate directly which loci are biotinylated. For this, the authors could extract and sonicate the genomic DNA and use streptavidin to enrich for biotinylated fragments. Strand-specific DNA sequencing could then be used to map the biotinylated loci.

In the cell culture medium, there were a certain amount of hemin from serum and a low dosage of biotin from the basal medium DMEM, which could not be avoid. Thus, these contaminated hemin and biotin would generate the background signals observed in the Non-label control samples. The biotinylated sites were specifically recognized by the recombinant Streptavidin monomer which further recruits Tn5 to the biotinylated sites with the help of Moon-tag. Different from the signals in the HEK293 samples, a much more robust HepG4-seq signals were observed in the mESC samples and the signals were also abolished in the non-label control samples. Thus, the relatively small signal-to-noise ratio in the HEK293 samples suggest the week abundance of endogenous G4s in the HEK293 cells. Thus, we politely disagree that hemin increased the non-specific recruitment of Th5. In addition, the CUT&Tag technology has been wildly demonstrated to have a much lower background, high signal-to-noise ratio and high sensitivity. Thus, we also politely disagree to replace the CUT&Tag with the traditional DNA library preparation method.

Figure 1H: No spike-in was added and the data are not quantitative. The number of replicates is unclear. 70000 extra peaks (10x) after inhibition of BLM or WRN seems enormous. These extra peaks should be better characterised: do they contain G4 motifs? Are they transcribed? etc...; again what kind of controls should be used here, in case the inhibition of BLP and WRN has a global impact on chromatin accessibility?

To quantitatively compare different samples, we have normalized all samples according their de-duplicated uniquely mapping reads numbers. Given that the inhibitors were dissolved in the DMSO, we used the DMSO as the control. Since the Tn5 were specifically recruited the biotinylated G4 sites through the recombinant Streptavidin monomer protein and the moon tag system, the chromatin accessibility will not affect the Tn5, which were normally observed in the ATAT-seq.

As suggested, we have analyzed the enriched motifs of the extra peaks induced by BLM or WRN inhibition and showed that the top enriched motifs are also G-rich in the supplementary Fig.1E. In addition, we analyzed the RNA-seq levels of genes-associated with these extra peaks. As shown in the figure below, the majority of these genes are actively transcribed.

**Author response image 1. sa4fig1:** 

Figure 2: The mutated version of HBD should have been used as a control. As shown clearly in PMID: 37819055, the HBD domain does interact in an unspecific manner with chromatin at low levels. As above, this might be enough to increase the local concentration of the Tn5 close to chromatin in the Cut&Tag approach and to cleave accessible sites close to TSS in an unspecific manner.

As shown in Fig.2B and Fig.4A, we have included the RNase treatment as the control and showed that the HBD-seq-identified R-loops signals are dramatically attenuated (Fig.2B) or almost completely abolished after the RNase treatment (Fig.4A). These data demonstrate the specificity of HBD-seq.

Figure 2: What fraction of the HEPG4-seq signal is sensitive to RNase treatment? The authors used a combination of RNase A and RNase H but previous data have shown that the RNase A treatment is sufficient to remove the HBD-seq signal (which means that it is not actually possible on this sole basis to claim or disclaim that the signals do correspond to genuine R-loops). Do the authors have evidence that the RNase H treatment alone does impact their HBD-seq or HEPG4-seq signals?

As shown in Fig.2B and Fig.4A, the HBD-seq-identified R-loops signals are all dramatically attenuated (Fig.2B) or almost completely abolished after the RNase treatment (Fig.4A). The specificity of HBD on recognizing R-loops has been carefully demonstrated in the previous study (PMID: 33597247). In this study, we used the same two copies of HBD (2xHBD) and replaced the GST tag to EGFP-V5 to reduce the possibility of variable high molecular-weight aggregates caused by GST tag. In addition, RNase H treatment has been shown to fail to completely abolish the CUT&Tag signals since a subset of DNA-RNA hybrids with high GC skew are partially resistant to RNase H (PMID: 32544226, 33597247). In consideration of the high GC skew of co-localized G4s and R-loops, we combined the RNase A and RNase H. We currently did not have the RNaseH alone samples.

Figure 3A: "RNA-seq analysis revealed that the RNA levels of co-localized G4s and R-loops-associated genes are significantly higher": the differences are not very convincing.

In the Figure 3A, we have performed the Mann-Whitney test to examine the significance in the revised manuscript. RNA levels of co-localized G4s and R-loops-associated genes are indeed significantly higher than all genes, G4s or R-loops- associated genes with the Mann-Whitney test p < 2.2E-16.

Figure 3B: the patterns for "G4" and "co-localised G4 and R-loop" are extremely similar, suggesting that nearly all G4s mapped here could also form R-loops. If this is the case, most of the HEPG4-seq signals should be sensitive to exogenous RNase H treatment or to the in vivo over-expression of RNase H1. This should be tested (see above).

The percentage of co-localized G4 and R-loop in G4 peaks is 80.3% (5,459 out of 6,799) in HEK293 cells and 72.0% (68,482 out of 95,128) in mESC cells, respectively. The co-localization does not mean that G4 and R-loop interact with each other. We have showed that only small proportion of co-localized G4s and R-loops displayed differential G4s and R-loops at the same time in the dhx9KO mESCs (Fig. 6D, Supplementary Fig. 3B), suggesting that the majority of co-localized G4s and R-loops do not interact with each other. Thus, we thought that it is not necessary to perform the RNase H test.

Figure 3C: there is no correlation between the FC of G4 and the FC of RNA; this is not really consistent with the idea that the stabilisation of G4 is the driver rather than a consequence of the transcriptional changes.

Given that the treatment of WRN or BLM inhibition induced a large mount of G4 accumulation (Fig.1H-I), we examined the transcription effect on genes associated with these accumulated G4s in Fig.3C. We indeed observed the effect of G4 accumulation on transcription of G4-associated genes. Given that G4 stabilization triggers the transcriptional changes, it does not mean that the transcriptional changes should be highly correlated with the increase levels of G4s. To our knowledge, we have not observed this type of connections in the previous studies.

l279: the overlap with H3K4me1 is really not convincing.

For all G4 peaks, the signals of H3K4me1 indeed exhibit a high background around the center of G4 peaks but we still could observe a clear peak in the center.

Figure 5C: it should be clearly indicated here that the authors compare Cut&Tag and ChIP data. The origin of the ChIP-seq data is also unclear and should be indicated.

Thank you for the suggestions. We have clarified this point.

For the ChIP data, we have described the origin of ChIP-seq data in the “Data availability” section as below: “The ChIP-seq data of histone markers and RNAP are openly available in GNomEx database (accession number 44R) (Wamstad et al., 2012).”

**Reviewer #2 (Recommendations For The Authors):**
(1) Figure 1A. An experimental condition lacking H2O2 (-H2O2) should be included.

We have added this control in Fig.1A

(2) Does RNAse H affect G4 profiles?

We have not tested the effect of RNase H on G4 forming. However, we have showed that only small proportion of co-localized G4s and R-loops displayed differential G4s and R-loops at the same time in the dhx9KO mESCs (Fig. 6D, Supplementary Fig. 3B), suggesting that the majority of co-localized G4s and R-loops do not interact with each other. Thus, we thought that it is not necessary to perform the RNase H test on G4. In addition, to treat cells wit RNase H, we have to permeabilize cells first to let RNase H enter the nuclei. If so, we will lose the pictures of endogenous G4s.

(3) Figure 2G. R-loops are detected upstream of the KPNB1 gene. What is this region? Is it transcribed?

We are so sorry to make a mistake when we prepared this figure. We have change it to the correct one in Fig. 2G. The R-loop is around the TSS of KPNB1. We also showed the RNA-seq data in this region in Author response image 2 below. This region is indeed transcribed.

**Author response image 2. sa4fig2:** 

(4) Did BLM and WRN inhibition specifically affect the expression of genes containing colocalized G4s and R-loops? Was the effect seen in other genes as well? Appropriate statistical analyses are needed.

In the Fig.3, we have shown that the accumulation of co-localized G4 and R-loops induced by the inhibition of BLM or WRN significantly caused the changes of genes (480 in BLM inhibition, 566 in WRN inhibition) containing these structures most of which are localized at the promoter-TSS regions. We indeed detected the effect in other genes as well. There were 918 and 1020 genes with significantly changes (padjust <0.05 & FC >= 2 or FC <= 0.5) in BLM and WRN inhibition, respectively.

(5) The claim that "The co-localized G4s and R-loops-mediated transcriptional regulation in HEK293 cells" (title of Figure 3) is not supported by the presented data. A causality link is not established in this study, which only reports correlations between G4s/R-loops and transcription regulation.

We politely disagree with this point. BLM and WRN are the best characterized DNA G4-resolving helicase (Fry and Loeb, 1999; Mendoza et al., 2016; Mohaghegh et al., 2001). Here, we used the selective small molecules to specifically inhibit their ATPase activity and observed dramatical induction of G4 accumulation. Notably, the accumulated G4s that trigger the transcriptional changes are mainly located at the promoter-TSS region. If the transcriptional changes trigger the G4 accumulations, we should not observe such a biased distribution and more accumulated G4s should be detected in the gene body.

(6) The effect of Dhx9 KO on colocalized G4s/R-loops and transcription is not clear. The suggestion that Dhx9 could regulate transcription by modulating G4s, R-loops, and co-localized G4s and R-loops is not supported by the presented data. Additional experiments and statistical analyses are needed to conclude the role of Dhx9 on colocalized G4s/Rloops and transcription.

Dhx9 has been extensively studied and reported to directly unwind R-loops and G4s or promote R-loop formation (PMID: 21561811, 30341290, 29742442, 32541651, 35905379, 34316718). Thus, it is not necessary to repeat these assays again. To understand the direct Dhx9-bound G4s and R-loops, we performed the Dhx9 CUT&Tag assay and analyzed the co-localization of Dhx9-binding sites and G4s or R-loops. 47,857 co-localized G4s and R-loops are directly bound by Dhx9 in the wild-type mESCs and 4,060 of them display significantly differential signals in absence of Dhx9, suggesting that redundant regulators exist as well. These data have clearly shown the roles of Dhx9 directly modulating the stabilities of G4s and R-loops. Furthermore, we showed that loss of Dhx9 caused 816 Dhx9 directly bound colocalized G4 and R-loop associated genes significantly differentially expressed, supporting the transcriptional regulation of Dhx9. We performed the differential analysis following the standard pipeline: DESeq2 for RNA-seq and DiffBind for HepG4-seq and HBD-seq. The statistical details have been described in the figure legends.

(7) The conclusion that Dhx9 regulates the self-renewal and differentiation capacities of mESCs is vague. Additional experiments are needed to elucidate the exact contribution of Dhx9.

In this study, we aimed to report new methods for capturing endogenous G4s and R-loops in living cells. In this study, we have shown that depletion of Dhx9 significantly attenuated the proliferation of the mESCs and also influenced the capacity of mESCs differentiation into three germline lineages during the EB assay. In addition, we showed that depletion of Dhx9 significantly reduced the protein levels of mESCs pluripotent markers Nanog and Lin28a. The comprehensive molecular mechanism of Dhx9 action is indeed not the focus of this study. We will work on it in the future studies. Thank you for the comments.

**Reviewer #3 (Recommendations For The Authors):**
The study on the involvement of native co-localized G4s and R-loops in transcriptional regulation further enriches the readers' understanding of genomic regulatory networks, and the functional dissection of Dhx9 also lays a good foundation for the study of the dynamic regulatory mechanisms of co-localized G4s and R-loops. Unfortunately, however, the authors lack a strong basis for questioning the widely used BG4 and S9.6 antibodies, and the co-localized G4s and R-loops sequencing data obtained by the developed and optimized method also lack parallel comparison with existing sequencing technologies, which cannot indicate that HepG4-seq and HBD-seq are more reliable and superior than BG4 and S9.6 antibody-based sequencing technologies. There are also some minor errors in the manuscript that need to be corrected.

Thank you for the constructive comments. We have added a new section (Comparisons of HepG4-seq and HBD-seq with previous methods) and a new figure 9 to parallelly compare our methods to other widely-used methods.

(1) This work mainly focuses on co-localized G4s and R-loops, but in the introduction section, the interplay between G4s and R-loops is only briefly mentioned. It is suggested that the importance of the interplay of G4s and R-loops for gene regulation should be further expanded to help readers better understand the significance of studying co-localized G4s and R-loops.

Thank you for the comments. The current studies about the interplay between G4s and R-loops are limited. We have summarized all we could find in the literatures.

(2) The authors mentioned that "a steady state equilibrium is generally set at low levels in living cells under physiological conditions (Miglietta et al., 2020) and thus the addition of high-affinity antibodies may pull the equilibrium towards folded states", in my understanding this is one of the important reasons why the authors optimized the G4s and R-loops detection assays, I wonder if there is a reliable basis for this statement. If there is, I suggest that the authors can supplement it in the manuscript.

The main reason we develop the new method is to develop an antibody-free method to label the endogenous G4s in living cells. We ever tried to capture endogenous G4s using the tet-on controlled BG4. Unfortunately, we found that even a short time induction of BG4 in living cells was toxic. The traditional antibody-based methos rely on permeabilizing cells first to let the antibodies enter the nuclei. In this case, it is easy to lost the physiological pictures of endogenous G4s. We will add more discussion about this point. For R-loops, we just further optimized the GST-2xHBD-mediated method to avoid the problem of GST-tag. GST-fusion proteins are prone to form variable high molecular-weight aggregates and these aggregates often undermine the reliability of the fusion proteins.

(3) Some questions about HepG4-seq:Is there a difference in hemin affinity for intramolecular G quadruplexes, interstrand G quadruplexes, and their different topologies? If so, does this bias affect the accuracy of sequencing results based on G4-hemin complexes?

Thank you for pointing out this issue. In the in vitro hemin-G4 induced self-biotinylation assay, parallel G4s exhibit higher peroxidase activities than anti-parallel G4s (PMID: 32329781). Thus, the dynamics of G4 conformation possibly affect the HepG4-seq signals. In the future, people may need to combine HepG4-seq and BG4s-eq to carefully explain the endogenous G4s. We have discussed this point in the revised version.

HepG4-seq is based on proximity labeling and peroxidase activity of the G4-hemin complex. The authors tested and confirmed that the addition of hemin and Bio-An in the experiment had no significant influences on sequencing results, but the effect of exogenous H2O2 treatment may also need to be taken into account since ROS can mediate the formation of G4s.

For HepG4-seq protocol, we only treat cells with H2O2 for one minute. Thus, we thought that the side effect of H2O2 treatment should be limited in such a short time.

(4) As we know, there have been at least two structure data of the S9.6 antibody very recently, and the questions about the specificity of the S9.6 antibody on RNA:DNA hybrids should be finished. The authors referred to (Hartono et al., 2018; Konig et al., 2017; Phillips et al., 2013) need to be updated, and the author's bias against S9.6 antibodies needs also to be changed. However, as the authors had questioned the specificity of the S9.6 antibody, they should compare in parallel with the data they have and the data generated by the widely used S9.6 antibody.

Thank you for the updating information about the structure data of S9.6 antibody. We politely disagree the specificity of the S9.6 antibody on RNA:DNA hybrids. The structural studies of S9.6 (PMID: 35347133, 35550870) used only one RNA:DNA hybrid to show the superior specificity of S9.6 on RNA:DNA hybrid than dsRNA and dsDNA. However, Fabian K. et al has reported that the binding affinities of S9.6 on RNA:DNA hybrid exhibits obvious sequence-dependent bias from null to nanomolar range (PMID: 28594954). We have included the comparison between S9.6-derived data and our HBD-seq data in the Fig.9 and the section “Comparisons of HepG4-seq and HBD-seq with previous methods”.

(5) It is hoped that the results of immunofluorescence experiments can be statistically analyzed.

We have performed the statistical analysis and included the data in the new figure.

(6) Some minor errors:Line 168, "G4-froming" should be "G4-forming";Figure 5E, the color of the "Repressed" average signal at the top of the HepG4-seq heatmap should be blue;Figure 7C, the abbreviation "Gloop" should be indicated in the text or in the figure caption.

Thank you for pointing out these issues. We are sorry for these mistakes. We have corrected them in the revised version.